# Women's experiences collecting and accessing water in Guatemala, Honduras, Kenya, and Zimbabwe: A mixed-methods investigation

Bethany A. Caruso[1]*, Thea Mink[1], Madeleine Patrick[1], Emily Ogutu[2], Cameron Dawkins[1], Olivia Bendit[1], Mahnoor Fatima[1], Ingrid Lustig[1], Alicia Macler[1], Jera White[1], Alondra Zamora[1], Alberto Emanuel Santos López[3], Héctor Salvador Peña Ramírez[3], Carlos Daniel Sic[3], Jorge Lemus Chávez[4], Sandra Antonio[5], Jazmina Nohemí Irías[5], Gladys Ramos[6], Everlyne Atandi[7], Peter Mwangi[7], Peter Koome[8], Rohin Otieno Onyango[8], Petronilla Andiba Otuya[8], Paul Ruto[8], Morris Chidavaenzi[9], Jammaine Jimu[9], Sithandekile Maphosa[9], Makaita Maworera[9], Munyaradzi Damson[10], Sheela S. Sinharoy[1]

1 Hubert Department of Global Health, Rollins School of Public Health, Emory University, Atlanta, Georgia, United States of America, 2 Gangarosa Department of Environmental Health, Rollins School of Public Health, Emory University, Atlanta, Georgia, United States of America, 3 World Vision Guatemala, Guatemala City, Guatemala, 4 Centro Universitario de Occidente, de la Universidad de San Carlos de Guatemala, Quetzaltenango, Guatemala, 5 World Vision Honduras, Tegucigalpa, Honduras, 6 Departamento de Ciencias Sociales, Universidad Nacional Autónoma de Honduras, Tegucigalpa, Honduras, 7 World Vision Kenya, Nairobi, Kenya, 8 School of Education and Social Sciences, St. Paul's University, Nairobi, Kenya, 9 World Vision Zimbabwe, Harare, Zimbabwe, 10 Datalyst Africa, Harare, Zimbabwe

* bcaruso@emory.edu

## Abstract

1.8 billion people live in households that collect water from sources off household premises, creating burdens that disproportionately affect women. A current and comprehensive investigation of women's water collection experiences is needed to understand their burden of this labor. This study used mixed methods, including (a) go-along, in-depth interviews (IDIs), (b) semi-structured observation, (c) activity-tracking smart watches, and (d) scales, to (1) understand women's practices, perspectives, and experiences going to water sources; (2) determine actual water collection time, distance, caloric expenditure, total elevation ascended, weight carried, and water volume collected; and (3) assess alignment of women's estimated and actual water journey times. Ninety-four women participated across four countries: Guatemala (n=22), Honduras (n=17), Kenya (n=22), Zimbabwe (n=33). Women reported accessing various sources depending on season and needs, faced risks due to terrain and animals, and experienced physical injury and mental burden. Experiences varied within and by country. The mean water journey time (including going to the source, activities at the source, and returning) was 82 minutes (range across entire sample: 13 minutes (Guatemala) – 287 minutes (Kenya)). The mean distance traveled was 3.5km (range: 0.2km (Guatemala, Honduras, Zimbabwe) - 15.8km

**Data availability statement:** All data and tools are publicly available via FigShare (https://doi.org/10.6084/m9.figshare.c.7640204).

**Funding:** This work was supported by World Vision US [SOW 34483]. A Creative Commons Attribution 4.0 Generic License has already been assigned to the Author Accepted Manuscript version that might arise from this submission. Author BAC was the recipient of the grant. The funder was involved in discussions related to the purposive selection of countries for data collection; World Vision country offices then helped to identify specific communities for data collection. The funder had no role in data collection, data analysis, data interpretation, or writing of the manuscript.

**Competing interests:** The authors have declared that no competing interests exist.

(Kenya)). Mean caloric expenditure was 231kcal (range: 36 (Guatemala) – 952 (Zimbabwe)). The mean volume of water collected was 16.1L. (range: 3.7L (Kenya) - 38.2L (Zimbabwe). Women also carried children, wet laundry, and other items resulting in heavy loads. The mean total weight of loads brought from water sources was 19.3 kg (range: 5.0 kg (Honduras) - 50.1 kg (Zimbabwe). Findings demonstrate how a lack of adequate and accessible water drains women of energy and time and poses risks to their well-being. Our findings reinforce the need to redouble efforts to improve water access in low-resource settings and rigorously measure the impacts of such efforts on women's lives.

## 1 Introduction

Water is critical for health [1] and sustaining life [2] and is a basic human right [3]; yet a large proportion of the global population still lacks access, creating burdens that disproportionately affect women and girls [4]. An estimated 1.8 billion people live in households where water is collected from sources located off household premises; in 70% of these households, women and girls are the primary water collectors (63% women and girls aged 15 and over; 7% girls younger than age 15) [4]. This unpaid labor and drudgery are driven by sociocultural norms that conscript women to carry out this time-consuming, arduous, and risk prone work [5,6]. These quantitative estimates of water collection responsibilities are vital for showing who bears this burden, but they do not capture what this burden entails. While thorough assessments of water collection time and energy requirements exist, they are outdated and primarily focused on East Africa [7,8]. Thus, a current, comprehensive, and more diverse investigation of water collection experiences, particularly among women, is needed to understand the gendered burden of this labor.

Research on how water fetching can impose disproportionate physical harms and cognitive demands on women is growing [9]. Water fetching has been linked to poor musculoskeletal health, broken limbs, lacerations, general pain, fatigue, perinatal health problems, stress, chest pain, headaches, and hair loss for those carrying on the head [10–17]. Research conducted in 21 low-income and middle-income countries found that 85% of respondents (6291/7401) reported experiencing a water-fetching-related injury. Among those, 72% were female [12]. Further, research from Odisha, India illuminated the cognitive burden of water collection and management among women who described the myriad factors they needed to weigh when determining where to access water, including household needs (e.g., drinking, cooking), water characteristics (e.g., taste), source features (e.g., distance), and season, among others [18].

The imperative to improve water access to save women's time and energy was argued nearly 40 years ago [19], and while water access has improved to varying extents, water collection continues to place substantial burdens on women's time and energy [20]. Specific understanding of the time required for water collection is improving, yet the impact of water collection on energy, beyond qualitative reports of

exhaustion, remains limited. Globally, women collectively spend an estimated 250 million hours per day collecting water, over three times the amount of time spent by men.[21] The longer the time needed to fetch water, the less likely that men are involved [22], further exacerbating the disproportionate burden placed on women. As such, women are more likely to fetch water when the work is intensified during dry seasons or climate-related droughts [23], which can require farther travel, longer queues, and elevated water extraction effort [24–26]. Moreover, as water collection times and distances increase, the volume of water collected decreases [7,19,27,28], meaning that women's increasing efforts likely yield water quantities that are insufficient to meet their needs.

Water collection time has been associated with individuals' education and economic opportunities and health [12,29–34], reinforcing the need for a more precise understanding of the amount of time spent collecting water. Specifically, increases in time for water fetching have been associated with increased risk of injury [12] and intimate partner violence [29]. Research in India found higher water-fetching times among children to be associated with lower school math, reading, and writing scores [30]. Conversely, improved water access has resulted in time savings, enabling leisure, social, and economic activities [31,32]. For example, the provision of piped water into households in Malawi resulted in a 77% reduction in time spent collecting water among women, 69% among girls, and 72% among boys, amounting to an estimated 3.8 hours of time savings per household per week, mostly for women and girls. Further, water use increased by 32%, including for domestic and productive uses [32]. Additionally, a decrease in roundtrip water collection time by 15 minutes in Sub-Saharan Africa was associated with improved child health outcomes [33], which may further reduce time women spend caring for sick children [34].

Better data on time, distance, and energy related to water collection are needed. Most data on water collection time are self-reported; other methods to measure time are sparse [17,35–37]. While self-report time data are critical for large-scale global monitoring efforts, more precise measures are needed for research [38]. A limited body of research has compared self-reported and measured water collection time with mixed results, finding self-reported time to be underestimated in some settings and overestimated in others [35–37,39]. Data are also limited on distances traveled to water sources [40]; among studies that report distance, most involved participants who traveled less than 1 kilometer [35,41–43]. Distance may be understudied because time is often used as a proxy [17,33,44]. However, time is an inaccurate substitute for measuring distance [37,45]; variability in terrain, elevation, weather, physical ability, health status and the load carried all influence the time it takes to traverse the same distance [46]. Similarly, GIS methods to determine Euclidean distance—from homes to the water source—have been found to underestimate the distances traveled compared to GPS methods that can measure the actual routes taken.[39] Finally, women likely expend considerable energy collecting water and doing water work, like laundry and watering animals and garden crops, at sources, yet little is known about this energy expenditure. Existing studies are limited by small sample sizes and use simulations to generate estimates [47,48]. Improving understanding of the burden of water collection—specifically the energy and time required and distance traveled—could inform policy, practice and investment related to water infrastructure, including decisions about whether to install piped water systems or where best to position community-level sources.

This study sought to comprehensively understand women's burden traveling to, performing work at, and returning from water sources in four countries: Guatemala, Honduras, Kenya, and Zimbabwe. It aimed to: [1] understand women's practices, perspectives, and experiences going to water sources; [2] determine actual water collection time, distance, caloric expenditure, total elevation ascended, work performed at source, weight carried, and water volume collected and transported; [3] assess alignment of women's estimated and actual water journey times.

## 2 Methods

### 2.1 Ethics

The study protocol was reviewed by the Institutional Review Board committee of Emory University, Atlanta, Georgia, USA, which determined that the research was exempt from further review and approval (IRB00005955). It was approved by

the Comité de Ética del Centro Universitario de Occidente, de la Universidad de San Carlos de Guatemala in Guatemala (Acta 1.23 C.E. DICUNOC); Universidad Nacional Autónoma de Honduras (CEIFCS-2023-P18) in Honduras; St. Paul's University - Institutional Scientific Ethics Committee (ERB No. 38), the National Commission for Science, Technology and Innovation (NACOSTI/P/23/27117) in Kenya; and the Medical Research Council of Zimbabwe (MRCZ/A/3054) in Zimbabwe. All participants provided informed consent. In Guatemala, Honduras, and Kenya, participants consented verbally, and enumerators signed the consent form to affirm. In Zimbabwe, participants provided signed consents.

## 2.2 Study design and motivation

This convergent parallel mixed methods study [49] used (a) go-along, in-depth interviews (IDIs), (b) semi-structured observation, (c) activity-tracking smart watches, and (d) scales to elucidate women's experiences going to water sources (Fig 1). Results are reported according to the Standards for Reporting Qualitative Research [50] (SRQR; See S1 Checklist).

   Research took place across four countries (Guatemala, Honduras, Kenya, and Zimbabwe) in communities where World Vision is delivering its *Strong Women, Strong World: Beyond Access* (SWSW) programming. The SWSW program seeks

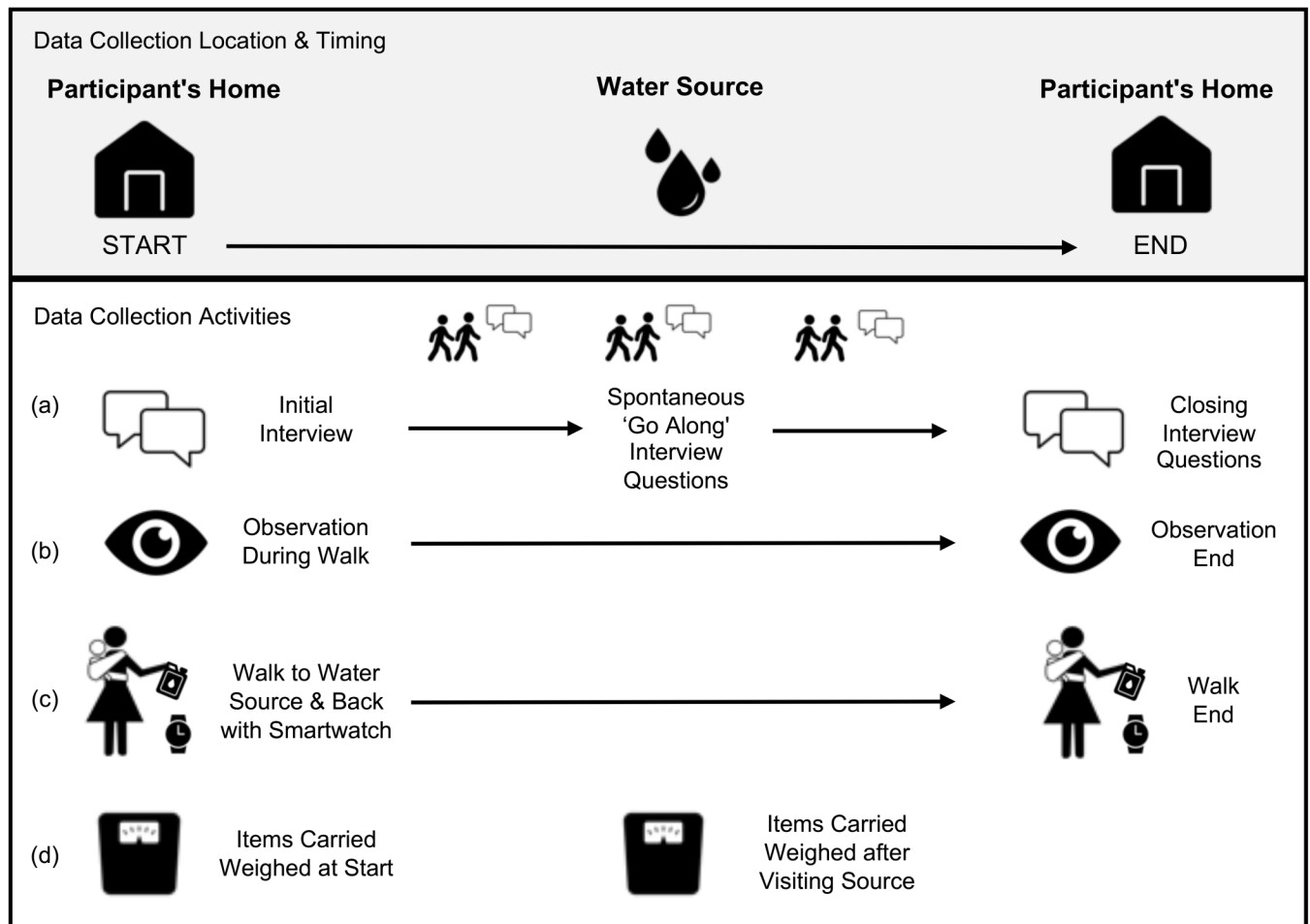

**Fig 1. Mixed-methods data collection activities to understand women's water collection experiences.**

to increase women's empowerment and improve women's well-being through an integrated intervention that includes water improvements and economic empowerment training and support. The SWSW program theorizes that by providing better water services, women will have increased access to water, saving them time and energy and enabling them to use water, time, and energy for other purposes, including economic activities. World Vision hypothesized that women in the SWSW program areas expend substantial time and energy to retrieve limited quantities of water. To test this hypothesis, Emory University worked with World Vision USA and local World Vision offices in Guatemala, Honduras, Kenya, and Zimbabwe to design the present study; Emory and the World Vision partners collectively agreed that research would be useful in all four countries where the program takes place to discern if and how experiences differed across these settings and to best inform the ongoing programs. We expected experiences to vary across countries based on known differences in geography, season, and water access. To carry out this research, each World Vision country office identified a local learning partner to contribute research expertise and lead data collection activities: Centro Universitario de Occidente de la Universidad de San Carlos de Guatemala, the National Autonomous University of Honduras, St. Paul's University in Kenya, and Datalyst Africa in Zimbabwe.

## 2.3  Community and participant sample size and eligibility, and setting specific details

We aimed to engage at least six communities in each country to capture diversity in geography and population. Communities in Guatemala, Honduras, Kenya, and Zimbabwe were purposively selected by World Vision country offices for their engagement in World Vision SWSW programming. Almost all communities selected had water interventions planned, but not yet completed. In Kenya, World Vision had just completed construction of a borehole in one community, which was used by participants in that community and from two neighboring communities. As such, participants from three of six engaged communities in Kenya already had access to the new source.

We aimed to engage at least three women per community for a total minimum sample of approximately 18 women per country. Samples of 9–17 participants have been deemed to be sufficient for qualitative research [51], and we used this as a guideline per country sample size. Where resources allowed, we sought to increase the number per community to increase variability of data captured with the smart watches. In all settings, we recruited and enrolled a convenience sample of individuals who were eligible to participate if they were: women, bore responsibility for going to water sources for household needs, lived in a community selected as eligible, and were age 18 or older. Specific details around participant recruitment varied slightly in each country, as described below.

Research in Guatemala took place in seven rural communities in the departments of Quetzaltenango and Totonicapán, mountainous regions in the southern part of the country during the rainy season. In Guatemala, 95% of the population (91% rural; 98% urban) has at least basic drinking water service, defined as an improved source requiring no more than 30 minutes roundtrip for collection [4]. Guatemala is a multicultural, multiethnic, and multilingual country; the majority of the population (56%) is of mixed indigenous-Spanish descent and 42% are Mayan. The research team deliberately recruited members of the Mayan population; women from two different Mayan linguistic groups (K'iche' and Mam) participated. Spanish is the official language of Guatemala and is spoken by 70% of the population; 8% of the Guatemalan population speak K'iche' and 4% speak Mam [52]. In Guatemala, individuals were invited to voluntarily participate through local community authorities, with whom the research team discussed the research. Some women were also recruited by the recommendation of other participating women. World Vision staff did not participate in recruitment.

Research in Honduras took place in six communities across two areas—one rural and one peri-urban—in the department of El Paraíso in eastern Honduras during the rainy season. The urban area is hilly and forested while the rural area is mountainous. In Honduras, 96% of the population (91% rural; >99% urban) has at least basic drinking water service [4]. Most (90%) of the population of Honduras is of mixed indigenous and European descent; 7% is indigenous. Spanish is the official language [53]. In Honduras, the local World Vision team introduced local community leaders (members of boards of trustees, water boards, churches) to the local research team members who explained the research project. The

local leaders brought the research team to neighborhoods within the community, in which the research team went door to door to explain the research and recruit eligible and interested participants. Neither World Vision staff nor the local leaders participated in recruitment.

Research in Kenya took place in six rural communities across Isiolo and Samburu Counties during the dry season. The areas are arid and semi-arid and characterized by drought and water scarcity. In Kenya, 63% of the population (53% rural; 86% urban) has at least a basic drinking water service [4]. Kenya is also a multicultural, multiethnic, and multilingual country. English and Swahili are both official languages and there are numerous indigenous languages [54]. In Kenya, the local research team approached village leaders in the study area to explain the study and seek their help identifying potential participants. The village leaders approached potential participants to gauge their interest in participating and relayed this information back to the research team. The team then visited interested individuals in person to confirm their eligibility, provide a detailed explanation of the study, and initiate the consent process. World Vision staff did not participate in recruitment.

Research in Zimbabwe took place in six rural communities in Gwanda and Nyanga districts. The districts are drought-prone, and data were collected during the dry season. Gwanda has flat, agricultural lands whereas Nyanga is forested, mountainous and has flat agricultural lands in the low-lying areas. In Zimbabwe, 62% of the population (48% rural; 93% urban) has at least basic drinking water service [4]. Shona (80.9%), Ndebele (11.5%), and English (0.3%) are the official languages [55]. Members of the Gwanda and Nyanga World Vision offices introduced the local research team to local village leaders, who provided lists of households from which women were selected, ensuring that they lived far from each other to increase variability. The research team visited women to explain the research and invite participation. Neither World Vision staff nor village leaders participated in recruitment.

Additional information regarding the ethical, cultural, and scientific considerations specific to inclusivity in global research is included in the S2 Checklist. Data will be made publicly available upon publication at FigShare.[56]

## 2.4 Data collection tools and procedures

Data collection took place from June-September 2023 (Guatemala: July 11-August 2; Honduras: July 5-August 5; Kenya: June 22-July 24; Zimbabwe: July 15-July 22 and September 22-September 30.

All participants provided answers to a short survey on demographic characteristics and water access (e.g., primary water source). Women were also asked two questions informed by the Demographic and Health Surveys (DHS) and Multiple Indicator Cluster Surveys (MICS) to get their estimates of water collection time: [1] "How long does it take for members of your household to go to the drinking water source, get water, and come back?" and [2] "How long does it take for members of your household to go to the water source for other uses, get water, and come back?"

To understand women's own perspectives about water collection and water work at sources, we carried out go-along in-depth interviews (See 'a' in Fig 1). Go-along interviews enable the interviewer and the participant to inhabit the space about which they are talking [57] and give participants active roles as guides through the space [58]. Because the environment can prompt discussion about what is observed and discussed, go-along interviews can produce richer insights than sedentary interviews [59]. This method has been used to examine women's water fetching experiences and environmental risks in an informal settlement in urban Malawi [15] and to explore experiences of water use, access, and insecurity in Western Kenya [60].

In our study, go-along interviews started in participant's homes, continued while going to and from the water source (during pauses in walking or at the source), and concluded back at participant's homes. Participants were asked to go to the water source they would normally be going to on that day and to carry out the activities they would typically carry out. The interview started while with participants were at their homes to build rapport before the journey and to not slow the participant or contribute to their exhaustion while walking. Example topics included those that pertained to participants' typical water-collection related routines, like where and when participants typically collect water, if they go in groups or alone, what activities they carry out either at the source or en route, what challenges they have come across, and what

they do with children while collecting. During the journey, the data collectors were guided to make specific observations and, as needed, ask clarifying questions based on observations, including if what they were seeing was typical. For example, data collectors were asked to observe and describe water containers used during the journey, the terrain traversed, the process of water collection, activities carried out in addition to water collection (e.g., breaks, feeding children), encounters with animals or other people, and if the participant experienced any injuries.

To capture additional information about experiences not gained through other methods, semi-structured observations were carried out during the journeys (See 'b' in Fig 1). Observation guides prompted team members to record information about, for example, the terrain, if women were accompanied, how water was extracted and transported home, and if women engaged in other activities on their journeys.

To determine total time, distance, elevation ascent, and caloric expenditure, participants wore Garmin Forerunner 255 activity-tracking watches (See 'c' in Fig 1). Watches were calibrated prior to the start of water collection using each participant's weight, height, and age to estimate caloric expenditure; caloric expenditure is reported in kcal. A data collection team member also wore a watch as a back-up, which was calibrated using the participant's weight, height, and age. Watches started recording when the participant left their home and finished upon return.

To determine the weight of items carried, Seca 876 scales were used (See 'd' in Fig 1). A data collection team member determined the weight of any items, such as a water container or a child, that each woman carried when she left home. The research team carried the scales to weigh anything acquired along the way. Items were re-weighed when women left the water source before beginning the return journey, such as water containers with water or wet laundry. If anything was difficult to place on the scale alone (e.g., clothing), team members weighed the woman while holding the item(s) and later subtracted her weight.

The protocol for go-along interviews, semi-structured observations, and research equipment and the tools for collecting participant demographic information and guiding go-along interviews and semi-structured observations are included as supplement (See S1 Text **and** S2 Text).

## 2.5  Data collection training

Emory team members trained the local research teams to carry out and supervise data collection. The five-day trainings provided background information about the SWSW program and guidance on ethical research conduct, principles of qualitative and quantitative research, data management, and how to use data collection tools and instruments (watches, scales, recorders). The local research teams also reviewed all tool translations to ensure consistent meaning and understanding across languages. In the last days of training, tools and procedures were piloted prior to data collection to ensure clarity, contextual appropriateness, and their usability. In Guatemala, Honduras, Kenya, and Zimbabwe, initial piloting was conducted among the enumerator teams to identify potential challenges in administering the tools. In Kenya and Zimbabwe, additional pilot testing was carried out with community members from areas adjacent to the main study sites. This process helped the teams identify questions that needed re-wording to strengthen clarity, required additional probing, or were poorly translated. Piloting with the community members also helped the teams in Kenya and Zimbabwe to estimate the duration of each data collection activity. Based on the pilot learning, the teams revised the tools and adjusted the data collection schedule accordingly. Data generated during the pilot phase was excluded from the final analysis.

## 2.6  Data collection and management

Two trained local research team members collected data from each participant. Typically, one team member explained and provided the watch to the participant, collected demographic data, and led the go-along, in-depth interview while the other team member took notes, led the semi-structured observation, and recorded data on all weighed items. Working in teams of two lessened the work burden and provided team members with a greater sense of security, particularly when journeying far in new locales.

Go-along, in-depth interview, observation, and demographic survey tools were translated into Spanish (Honduras and Guatemala), Samburu (Kenya), and Ndebele and Shona (Zimbabwe). Data collection team members in Guatemala who were native K'iche' or Mam speakers verbally translated tools from Spanish into K'iche' and Mam, as these languages are not typically written.

Digital voice recorders were used to record go-along interviews. Audio recordings in K'iche', Mam, and Spanish were first transcribed into Spanish by trained members of the local research teams and then translated into English by a hired translator. Audio recordings in Samburu, Ndebele, and Shona were simultaneously transcribed and translated into English by trained team members.

Demographic and observation data were recorded on paper forms. Trained team members in Kenya and Zimbabwe typed up observation notes in English. In Guatemala and Honduras, observation notes were typed up in Spanish and then translated into English by a hired translator. Demographic and select numeric or short answer data from observation sheets, which included weight data, were double entered into a database (by different members of the Emory team) and then compared for accuracy. Discordant data were checked against scanned files and corrected.

Data from the watches were synced to the 'Garmin Connect' web platform. No names were linked to the watches; each watch had an ID that the team used for tracking which watches were used for each journey ('observation'). On the online platform, each observation was renamed with a unique participant ID; this ID did not contain any identifiable information. While some participant information (weight, height, age) was used to calibrate the watch in advance of the observed journey, this information was updated for each new journey observed and did not remain on the watch. Data were then downloaded and exported from the Garmin web platform as CSV and TCX files and uploaded onto a secure OneDrive folder. Data were deleted from the watches after it was uploaded.

All data, including recordings and scanned documents (e.g., surveys, notes), were de-identified, uploaded to secure OneDrive folders, and stored using unique identifiers that linked all data for each participant.

## 2.7 Data analysis

To learn about women's practices, experiences, and perceptions of water collection, and the environments and factors that influence them, (aim 1; activities 'a' and 'b' in Fig 1), we conducted a rapid thematic analysis of all transcripts from go-along interviews and observation notes. The themes were chosen a priori based on the prompts in the data collection tools. Themes selected included: timing of trips to water sources, number of trips to water sources, methods used for water carriage, terrain encountered, perceived risks and experienced harms due to water collection/visiting water sources, the accompaniment of children when going to water sources and why (e.g., need to care for children, children assisted), additional work carried out at or on the way to or from water sources, perceived quality of water and any steps taken to treat water. Team member and co-author MP used Excel to organize data by creating four country-specific matrices that present themes by community. MP read each transcript and all observation notes, and any information relevant to a theme was summarized in the appropriate theme cell for community.

After data summarization by community for each country was complete, the matrices were carefully reviewed. MP and BC discussed the matrices and made iterative updates in three cases to combine themes that were closely related for clarity. Specifically, the themes "Number of trips" and "timing of trips" were combined because the timing of trips was dependent in part on the number of trips. "Terrain" and "risk" were combined because the risks identified by participants were closely related to the terrain and read as redundant when separated. "Perceived water quality" and "water treatment" were combined because participants typically spoke of the two themes together and because we had limited data on water treatment. After each country matrix was adjusted, MP created summaries for each theme by country. These summaries were then used to create one table with a country profile for each theme.

To determine actual water collection time, distance, caloric expenditure, total elevation ascent, and water volume and weight carried for the entire water journey (both to and from water sources) (aim 2; activities 'c' and 'd' in Fig 1), we

calculated descriptive statistics of watch and weight data. The volume of water collected was derived from the weight data; the weight of containers was subtracted from the total weight of the water-filled containers to determine weight of the water alone. The weight of water was then converted to volume (1 kg = 1L). We generated graphs to visualize trends related to participant journeys, with attention to time, distance, caloric expenditure, water volume, and the source visited (improved or unimproved).

Time, distance, total elevation ascent, and caloric expenditure data were manipulated for select participants in each country based on identified data collection challenges, enabling us to retain and use data collected. Specifically, in Honduras, one water journey was only recorded one-way. For this observation, time and distance were doubled to estimate a round-trip. In Kenya, data for two water journeys are the mean of the participant and enumerator watches as these were not properly distinguished during data collection (individual observations were similar). Watch data from data collectors were used as proxies for two participants (Guatemala, Zimbabwe) because participant water journeys were accidentally not recorded.

To determine if women's estimated and actual water collection times were aligned (aim 3), we calculated descriptive statistics (mean, range, SD) of the watch-measured times and women's estimates of water collection time, specifically among the subset that collected water. We calculated the difference between estimated and actual times and identified the number and proportion of observations by country and type (drinking, other uses) that were aligned and that were discordant. We leveraged the go-along interview and observation data to determine and classify women's intended use of the water they were collecting. Water was classified as for 'drinking' if a participant indicated that any of the water would be used for drinking, even if they also would use some of the water for other purposes. Water was classified as for 'other uses' (e.g., bathing, washing dishes) if women did not explicitly mention using any of it for drinking. (We acknowledge that the WHO/UNICEF Joint Monitoring Programme for Water Supply, Sanitation, and Hygiene (JMP) considers 'drinking water' to include 'drinking, cooking, personal hygiene and other domestic uses [61]. However, we separated as described because we expect participants responded to what was asked of them, not to otherwise determined categories (See section 2.3).

Quantitative analyses were done in Stata (version 18.0), Microsoft Excel, and R (v4.2.1); qualitative analyses were facilitated using Microsoft Excel.

## 2.8 Risk mitigation

Like all research, we acknowledge that this research could pose some unintentional risks. Specifically, we anticipated that the wearing of smart watches could cause conflict within families or communities or impose a safety risk to participants. We took steps to mitigate these risks by making it clear within communities that devices were not a gift but were to be returned to the research team. We clearly explained risks during consents so that women could opt out if they perceived that temporarily possessing a watch would create risk. Additionally, research can pose risks to those who collect data. To ensure the safety of our data collection team members, we hired and trained individuals from the broader research areas. We never had only one research team member accompany women on water journeys; water journeys were completed in teams of two. Finally, the research team intentionally worked with World Vision teams to ensure that research would not take place in areas with high perceived risks of banditry, where participants and research team members may be targets.

## 2.9 Reflexivity statement

This manuscript includes authors from the United States, Guatemala, Honduras, Kenya, and Zimbabwe. The authors are both men and women and represent a range of experiences, including in program delivery and in research (both globally and specific to each country). Members of the Emory team were engaged by World Vision US to a learning partner for the *Strong Women Strong World Program*. In that capacity, members of the Emory team facilitated meetings with each World Vision country team to identify their research priorities; assessing water collection was a priority across country teams.

Acknowledging the distance they should keep from the research given ongoing programs, World Vision country teams engaged local learning partners to lead data collection. Still World Vision country team members helped identify communities within which to collect data. Both members of the World Vision country teams and members of local learning partner teams leant their expertise about the country and communities by providing feedback on and strengthening the tools and research protocols that Emory drafted. Members of the Emory team led trainings in each country. Research participants only had direct contact with members of the local learning partner teams, who were all from the country they worked in and were fluent in the local languages. Analyses of data were led by the Emory team members who have extensive training in qualitative and quantitative analyses and the least assumptions about the data from each country. Once data analyses were completed, Emory led online workshops with members of the World Vision country and learning partner teams to discuss and raise questions about the findings (one in English with Kenya and Zimbabwe; one in Spanish with Guatemala and Honduras). In these meetings, Emory presented data and all present engaged in discussion about the data and its implications, which informed the current manuscript, particularly the discussion.

## 3 Results

### 3.1 Participant information

Ninety-four women from 25 communities in four countries participated in data collection (Table 1). For 93 women, a single journey to a single water source was recorded. One woman in Guatemala participated in two journeys to capture that participant's different water sources.

Across all settings, the mean participant age was 37.1 years (SD = 13.7). The largest proportions of participants in Guatemala (12; 55%) and Honduras (11; 65%) were unmarried with a partner or single; the majority in Kenya (17; 77%) and Zimbabwe (23; 70%) were married. The average household size was 5.8 members (SD = 2.9). The greatest proportion to never attend school was in Kenya (20; 91%) and the greatest proportion to engage in economic activities was in Guatemala (20; 91%).

### 3.2 Reported and observed water collection practices and experiences

**Sources.** Based on the demographic survey, women reported a variety of sources from which they would primarily collect water for drinking and 'other uses.' Overall, most participants, including all from Kenya and Zimbabwe, reported that their primary drinking water source was located outside of their dwelling, yard, or plot (83; 89%) and almost half (42; 46%) indicated that they collected drinking water from a source that we classified as unimproved (as per JMP definition [61]). However, there was variation between countries. Over half of the participants in Guatemala (15; 71%), Honduras (11; 65%), and Zimbabwe (19; 58%) reported collecting drinking water from a source considered improved, compared to just 24% [5] in Kenya. Surface water was the most reported unimproved drinking water source in Guatemala (5; 24%), Honduras (4; 24%) and Zimbabwe (13; 39%). In Kenya, most participants reported that they typically collect drinking water from unprotected dug wells (15; 71%). On average, women reported collecting water almost daily (5.6 days/week); though the mean number of reported collection days was lower in Honduras compared to the other countries (Guatemala 6.4 days; Honduras 4.1 days; Kenya 5.7 days; Zimbabwe 5.9 days) (See Table 2b and section 'Frequency of trips' for further elaboration from participants about how frequently they collect water and why).

While all women in Kenya reported that they would use the same water sources for drinking and 'other uses', sizable proportions of women in Guatemala (91%), Honduras (41%), and Zimbabwe (46%) said they visited different sources for drinking and 'other uses.' The proportions of women using surface water for 'other uses' were greater than the proportions using it for drinking in both Guatemala (+21 percentage points) and Zimbabwe (+23 percentage points). No women reported using rainwater for drinking, but approximately one third reported using it for 'other uses' in both Guatemala (6; 33%) and Honduras (5; 29%).

**Table 1.** Demographic and water characteristics of water journey participants, by country.

| | Country | | | | | | | | | |
|---|---|---|---|---|---|---|---|---|---|---|
| | Guatemala (n = 22; 7 communities) | | Honduras (n = 17; 6 communities) | | Kenya (n = 22; 6 communities) | | Zimbabwe (n = 33; 6 communities) | | Total (n = 94; 25 communities) | |
| **Age** (mean, SD) | 36.2 | 15.4 | 38.9 | 14.5 | 30.2 | 10.1 | 41.3 | 12.8 | 37.1 | 13.7 |
| **Household size[1]** (mean, SD) | 6.5 | 3.5 | 4.2 | 2.2 | 6 | 2.3 | 5.9 | 2.8 | 5.8 | 2.9 |
| **Marital status** (n, %) | | | | | | | | | | |
| Single, never married | 3 | 13.6 | 4 | 23.5 | 1 | 4.5 | 1 | 3.0 | 9 | 9.6 |
| Unmarried, but have partner | 9 | 40.9 | 7 | 41.2 | 1 | 4.5 | 0 | 0.0 | 17 | 18.1 |
| Married | 8 | 36.4 | 3 | 17.6 | 17 | 77.3 | 23 | 69.7 | 51 | 54.3 |
| Separated | 2 | 9.1 | 1 | 5.9 | 1 | 4.5 | 3 | 9.1 | 7 | 7.4 |
| Divorced | 0 | 0.0 | 0 | 0.0 | 0 | 0.0 | 2 | 6.1 | 2 | 2.1 |
| Widowed | 0 | 0.0 | 2 | 11.8 | 2 | 9.1 | 4 | 12.1 | 8 | 8.5 |
| **Completed schooling[2]** (n, %) | | | | | | | | | | |
| Never attended school | 4 | 18.2 | 4 | 25.0 | 20 | 90.9 | 0 | 0.0 | 28 | 30.4 |
| Some primary | 9 | 40.9 | 6 | 37.5 | 0 | 0.0 | 8 | 25.0 | 23 | 25.0 |
| Primary | 7 | 31.8 | 5 | 31.3 | 1 | 4.5 | 11 | 34.4 | 24 | 26.1 |
| Secondary | 2 | 9.1 | 1 | 6.3 | 0 | 0.0 | 13 | 40.6 | 16 | 17.4 |
| Above secondary | 0 | 0.0 | 0 | 0.0 | 1 | 4.5 | 0 | 0.0 | 1 | 1.1 |
| **Engaged in economic activities** (n, %) | 20 | 90.9 | 6 | 35.3 | 6 | 27.3 | 25 | 75.8 | 57 | 60.6 |
| **Primary source of drinking water[3]** (n, %) | | | | | | | | | | |
| *Improved Sources* | *15* | *71.4* | *11* | *64.7* | *5* | *23.8* | *19* | *57.6* | *50* | *54.3* |
| Piped water | 8 | 38.1 | 7 | 41.2 | 0 | 0.0 | 4 | 12.1 | 19 | 20.7 |
| Tube well/borehole | 0 | 0.0 | 0 | 0.0 | 3 | 14.3 | 14 | 42.4 | 17 | 18.5 |
| Protected dug well | 7 | 33.3 | 1 | 5.9 | 2 | 9.5 | 0 | 0.0 | 10 | 10.9 |
| Protected spring | 0 | 0.0 | 0 | 0.0 | 0 | 0.0 | 1 | 3.0 | 1 | 1.1 |
| Bottled water | 0 | 0.0 | 3 | 17.7 | 0 | 0.0 | 0 | 0.0 | 3 | 3.3 |
| *Unimproved Sources* | *6* | *28.6* | *6* | *35.3* | *16* | *76.2* | *14* | *42.4* | *42* | *45.6* |
| Unprotected dug well | 0 | 0.0 | 2 | 11.8 | 15 | 71.4 | 1 | 3.0 | 18 | 19.6 |
| Unprotected spring | 1 | 4.8 | 0 | 0.0 | 1 | 4.8 | 0 | 0.0 | 2 | 2.2 |
| Surface water | 5 | 23.8 | 4 | 23.5 | 0 | 0.0 | 13 | 39.4 | 22 | 23.9 |
| Other (unspecified) | 1 | 4.5 | 0 | 0.0 | 0 | 0.0 | 0 | 0.0 | 1 | 1.1 |
| **Location of drinking water source[4]** (n, %) | | | | | | | | | | |
| In dwelling | 3 | 13.6 | 1 | 5.9 | 0 | 0.0 | 0 | 0.0 | 4 | 4.3 |
| In yard/plot | 4 | 18.2 | 2 | 11.8 | 0 | 0.0 | 0 | 0.0 | 6 | 6.5 |
| Elsewhere (beyond yard/plot) | 15 | 68.2 | 14 | 82.4 | 21 | 100.0 | 33 | 100.0 | 83 | 89.2 |
| **Days/week needed to collect water[5]** | | | | | | | | | | |
| Mean, SD | 6.4 | 1.5 | 4.1 | 2.5 | 5.7 | 1.9 | 5.9 | 2.0 | 5.6 | 2.1 |
| Range | (2 –7 ) | | (1 –7) | | (2 –7 ) | | (1 –7) | | (1 –7 ) | |
| **Primary source of water for other uses[6]** (n, %) | | | | | | | | | | |
| *Improved Sources* | *10* | *55.6* | *12* | *70.6* | *5* | *23.8* | *10* | *31.2* | *37* | *42.1* |
| Piped water | 2 | 11.1 | 6 | 35.3 | 0 | 0.0 | 1 | 3.1 | 9 | 10.2 |
| Tube well/borehole | 0 | 0.0 | 0 | 0.0 | 3 | 14.3 | 9 | 28.1 | 12 | 13.6 |
| Protected dug well | 2 | 11.1 | 1 | 5.9 | 2 | 9.5 | 0 | 0.0 | 5 | 5.7 |
| Rainwater | 6 | 33.3 | 5 | 29.4 | 0 | 0.0 | 0 | 0.0 | 11 | 12.5 |
| *Unimproved Sources* | *8* | *44.4* | *5* | *29.4* | *16* | *76.2* | *22* | *68.8* | *51* | *58.0* |
| Unprotected dug well | 0 | 0.0 | 2 | 11.8 | 15 | 71.4 | 2 | 6.3 | 19 | 21.6 |
| Unprotected spring | 0 | 0.0 | 0 | 0.0 | 1 | 4.8 | 0 | 0.0 | 1 | 1.1 |

*(Continued)*

| | Country | | | | | | | | |
|---|---|---|---|---|---|---|---|---|---|
| | Guatemala (n = 22; 7 communities) | | Honduras (n = 17; 6 communities) | | Kenya (n = 22; 6 communities) | | Zimbabwe (n = 33; 6 communities) | | Total (n = 94; 25 communities) |
| Surface water | 8 | 44.4 | 3 | 17.6 | 0 | 0.0 | 20 | 62.5 | 31 | 35.2 |
| Other (unspecified) | 4 | 18.2 | 0 | 0.0 | 0 | 0.0 | 1 | 3.0 | 5 | 5.4 |
| Respondents with source for 'other uses' that is different than primary drinking water source[7] | | | | | | | | | |
| (n, %) | 20 | 90.9 | 7 | 41.2 | 0 | 0.0 | 15 | 45.5 | 42 | 45.2 |

[1]*Household size:* 1 missing (Zimbabwe)

[2]*Completed schooling:* 1 missing (Honduras); 1 missing (Zimbabwe)

[3]*Primary source of drinking water:* 1 missing (Kenya); Sources noted as 'other' were not classified as either improved or unimproved and were excluded from percentage totals

[4]*Location of primary drinking water source:* 1 missing (Kenya)

[5]*Days in a week needed to collect water:* 4 do not know (Guatemala); 1 missing (Kenya)

[6]*Primary source of water for other uses:* 1 missing (Kenya); Sources noted as 'other' were not classified as either improved or unimproved and were excluded from percentage totals

[7]*Different primary source types for drinking and 'other uses':* 1 missing (Kenya)

On the days they were followed, women visited improved and unimproved sources. In each of the three communities in Kenya where participants had access to a recently installed borehole, water was not always available. In each of these communities, two participants went to the new borehole and two went to an existing unprotected source. Participants also described other sources they would frequent depending on need or season (discussion of season was spontaneous, not probed). For example, participants from all countries reported collecting rainwater during the rainy season. Women from some communities discussed how they sometimes got water from family or neighbors (Honduras, Guatemala) or by paying someone, typically men (Zimbabwe). Table 2 provides thematic summaries of the qualitative data by country; S1 Table provides country-specific summaries by community.

**Activities at the source.** The primary reason for most (83; 87%) participants to visit water sources was to collect water. Among those, four women (2 Guatemala, 2 Kenya) also brought clothing to wash (Table 3). Ten women (9 Guatemala, 1 Kenya) did not collect water; they visited sources to wash clothes, which they brought with them.

The methods women used to extract water varied depending on the source. In Guatemala women used buckets or 'tinajas' (clay pots) to pull water from a well or filled containers by dipping them into rivers. Some women in both Guatemala and Honduras used hoses to fill containers. Other methods for extracting water were more arduous. Women in both Honduras and Zimbabwe used hand pumps, which women said required strength or as noted in Zimbabwe, assistance from other women also present at the sources. In Kenya and Zimbabwe, women dug for water at riverbeds. The process required them to dig deep enough to reach water, wait for water to pool, and then remove it with small vessels until it ran clear enough to be deemed suitable for collection. Women in Kenya said they sometimes needed to dig so deep during the dry season that they would be fully inside the holes.

In addition to water collection or washing clothing, many women carry out other activities at or on the way to and/or from water points. For example, at least one woman in each country was simultaneously caring for a child, carrying them on their back or front; in Guatemala and Honduras, at least one woman in each community reported that children regularly accompany them to water sources. Women noted older children occasionally accompany them and help carry water if strong enough. While at water points women also washed the clothing they were wearing (Guatemala, Kenya, Zimbabwe) and eating utensils (Guatemala); bathed themselves or children (Honduras, Kenya); and tended to and watered small vegetable gardens near the source (Zimbabwe). While journeying to or from water points, women discussed or were

PLOS Global Public Health | https://doi.org/10.1371/journal.pgph.0004355   December 17, 2025

**Table 2. a. Thematic descriptions of women's water collection experiences and practices, by country. b. Thematic descriptions of women's water collection experiences and practices, by country. c. Thematic descriptions of women's water collection experiences and practices, by country. d. Thematic descriptions of women's water collection experiences and practices, by country.**

a.

| Country & Theme | Guatemala (7 communities, 22 Participants) | Honduras (6 communities, 17 Participants) | Kenya (6 communities, 21 Participants) | Zimbabwe (6 communities, 33 Participants) |
|---|---|---|---|---|
| Water Sources Visited During Data Collection | Across communities, participants went to a variety of different sources, including rivers, wells, springs, and public tanks. | Across communities, participants went to a variety of different sources, including springs, wells, public and private taps, and rivers. | Across all communities, women collected water from an unprotected well, which they would dig in the riverbed. Some women in 3 communities (K4, K5, & K6) collected from a new borehole. Women in communities K1 & K4 reported some water pooling in rocks at the top of a hill as another source. | Across 5 of 6 communities (Z1, Z2, Z3, Z5, & Z6), women visited unprotected dug wells at water sources. In 2 communities (Z1 & Z3), participants visited dams. In 4 communities (Z3, Z4, Z5, & Z6), women also visited improved water sources such as taps (Z4 & Z6) or boreholes (Z3, Z4, Z5, & Z6). |
| Other Water Sources Used by Participants, But Not During Data Collection | Women said rivers or springs are available in all communities. In 4 communities (G2, G3, G4, & G6), women reported that some of their drinking water was provided for them for free by family members or neighbors who had household access to water. In 5 communities (G3, G4, G5, G6, & G7), there is a community water source such as a tank or a well. In all but 1 community (G6), women discussed rainwater collection during the winter rainy season. | Women in all communities reported accessing a natural water source, such as a river or spring. Rainwater harvesting was reported in 3 communities (H1, H3, & H5) to collect more water during the winter months. In 3 communities (H2, H3, & H5), women report that they may access drinking water from family members or neighbors with household water connections. | Women reported using alternative water sources during the rainy season, specifically surface water collection (e.g., small dams, gullies, small streams). Women in community K2 noted collecting rainwater, and women in community K1 described using an unprotected spring. Most women in communities with borehole access reported that they would access water via dug well at the riverbed for some purposes. In one community (K4) a new borehole was noted as an alternative source that was a great help, but the participant chose to collect water at the riverbed for an undisclosed reason. | Participants in 3 communities (Z3, Z4 & Z6) reported access to improved water sources: a borehole (in Z4 & Z6) and a tap at a school (Z3). A participant in community Z3 chose to collect water from the dam because the water in the borehole was running dry. Women in 5 of 6 communities (Z1, Z2, Z3, Z5, & Z6) reported using additional water sources in rainy season, such as rainwater harvesting from rooftops (Z1, Z3, Z5, & Z6), or surface waters such as streams or "puddles" (Z1, Z2, Z3, Z5, Z6). One participant in community Z3 occasionally paid someone for water. |
| Method of collection | Limited information was recorded about methods of collection at the source. In community G1, the woman who was collecting water (others were washing clothing), drew water from a well using a bucket and in community G2, a woman filled her containers directly from the river. In community G3, two participants extracted water from the source using a hose. (In communities G4, G5, & G6, the methods of collection were not recorded.) | Limited information was recorded about methods of collection at the source. In 2 communities (H2 & H3), water was piped from a source and women collected water through the hose. In community 6, one participant had to manually pump to get water. | Across all communities, women who collected water at the riverbed dug the well at the river, scooping out and throwing away the brown, sandy water initially seeping in. They then collected the clearer water after the well re-filled. In community K6, women said that three people needed to get inside the dug well to get the water to clarify the depth of the hole. In 2 communities (K1 & K2), women described using water pans to collect rainwater in the rainy season. | In 5 of 6 communities (Z1, Z2, Z3, Z5, & Z6), women dug a hole in or near a riverbed (or used a hole that had been dug by others) to collect water. In 3 communities (Z3, Z5, & Z6), participants that used the borehole reported needing the help of others to pump water. (In community Z4, the method of water collection was not noted.) |

(Continued)

Table 2. (Continued)

| Country & Theme | Guatemala (7 communities, 22 Participants) | Honduras (6 communities, 17 Participants) | Kenya (6 communities, 21 Participants) | Zimbabwe (6 communities, 33 Participants) |
|---|---|---|---|---|
| **b. Number and timing of trips** | Participants reported collecting water between 1 and 20 times a day, depending on how much water was needed and the availability of water at the source. In all communities, women reported leaving the house to fetch water less often in the rainy season. In community G4, one participant was careful to only collect once a day because she pays to travel to a water source. | The number of times water is collected varies across participants, communities, and seasons, from water collection every two weeks (community H6) to water collection over 10 times in a day (community H4). Women in 5 communities (H1, H2, H3, H4, H6) reported going more often in the summer dry season than the winter rainy season. In one community (H5), a participant reported going half as often in the summer because there is less water available to collect. When women go during the day depends on weather, needs, and other responsibilities. In 3 communities (H4, H5, H6), women said they time their water collection to avoid the heat of the sun. In 1 community (H3), a participant reported avoiding the water source in the afternoon because of mosquitoes. | Most participants who go to the riverbed go only once a day. Participants in 3 communities (K3, K5, K6) noted only going once per day because of the distance. Those in 3 different communities (K1, K2, K4) said they may go twice to the riverbed if needed, but those in community 4 clarified that it is not possible to go twice in the dry season. Women in two communities (K5, K6) said they sometimes go to the borehole twice in a day, but said going twice is not always possible because the borehole is not open every day, there are long lines, and the hours are limited. Women in two communities (K2, K5) said they may go more often in the rainy season to smaller, closer sources. Women across all communities discussed collecting water in the morning because of the heat in the afternoon and, as those in community K5 noted, it could be dark by the time they get home if they start later in the day. | The number of times women would collect water per day varied. In 2 communities (Z1, Z2), women who got water via sand abstraction reported collecting 1–3 times a day; one woman collected up to 5 times. Participants in 3 communities (Z4, Z5, Z6) collected water from dug wells 2–3 times per day and made trips to a borehole or tap. The number of trips depended on the quantity of water they could collect per trip, which would increase if they had use of a wheelbarrow or scotch cart (Z1, Z2, Z3, Z6). In 4 communities (Z1, Z2, Z5, Z6), women said they decreased the number of trips in the rainy season because they had closer sources then. In 4 communities, women said they would collect early in the morning to avoid long queues at the river (Z1, Z6) or borehole (Z3, Z4). Weather (Z2), borehole hours (Z3), events (e.g., funerals) (Z3), and visitors, who participants did not want to leave at their house (Z6) also influenced timing. |
| **Method of carriage** | A variety of tactics for carrying water were observed or described. In 3 communities (G4, G5, G6), at least one participant reported transporting water or laundry in a vehicle, for which participants occasionally paid (G4, G6). In 2 communities (G2, G4), women balanced water containers ('tinajas') on their heads, cushioned by a rolled-up blanket ('yagual'). A participant from community G1 noted that her husband would help her carry water, and a participant in community G5 said that if she needs to carry more than one container, she ties them together with a rope and pulls them. A participant from community G1 who did laundry carried the laundry in a large sheet. | Women across all communities carried water in buckets by their handles. In 2 communities (H2, H4), some women carried water on their heads. | Women across all communities collected water in jerricans that hold 20 liters, 10 liters, or, in 2 communities (K4, K5), 5 liters of water. Women would carry these on their backs or drag or roll them behind them. Decisions around which jerrican to use were based on women's strength or exhaustion, and if children were being carried. If older children were accompanying their mothers, they might assist by carrying small (5L) jerricans, or by rolling a jerrican. In 3 communities (K1, K2, K4), women said donkeys used to carry water, but that they died in the drought. | Participants in all communities carry water in buckets on their heads. Some participants in 4 communities (Z1, Z2, Z3, Z6) reported occasionally having access to wheelbarrows or scotch carts to facilitate water carriage, which enabled them to collect more water. |

*(Continued)*

**Table 2.** (Continued)

c.

| Country & Theme | Guatemala (7 communities, 22 Participants) | Honduras (6 communities, 17 Participants) | Kenya (6 communities, 21 Participants) | Zimbabwe (6 communities, 33 Participants) |
|---|---|---|---|---|
| **Terrain and risks** | In 6 of 7 communities (G1, G2, G3, G4, G6, G7), women described their path to water sources as steep and slippery, especially during the rainy season. Women in 3 communities (G1, G2, G7), reported that either they or a family member had slipped and fallen while collecting water. In 2 communities (G2, G6), women stated that in the rain, rivers would overflow and become dangerous. In 3 communities (G1, G2, G3), partic- ipants said that there were snakes on the pathway to the water source; dogs and people were also noted risks (G5). | Women in all communities reported routes to the water source to be slippery or become more treacherous in the rain. Participants in 2 commu- nities (H2, H3) reported snakes along the path and those in another (H3) reported coyotes and mosquitoes. | The terrain was mostly flat in several communities but was hilly in 2 (K4, K6). Women across all communities traversed gullies on the way to and from water collection points, which could pose challenges. Women from all com- munities also reported wild elephants to be a risk during water collection, and some also reported risks from leopards and other people. | In all communities, the routes to the water sources are hilly or mountainous, with loose stones. In 5 of 6 communities (Z1, Z3, Z4, Z5, Z6), there are sandy stretches that can be hot and difficult to walk on while carrying water or pushing wheelbarrows. In 4 communities (Z2, Z3, Z5, Z6), there are prickly or thorny trees and bushes on the pathway. Participants in 2 communities (Z4, Z5) said motor- bikes go by quickly making it dangerous. In 3 communities (Z2, Z3, Z6), women reported there to be dangerous people, including those who sexually harassed or assaulted women (Z2, Z6) or children (Z3), or traditional healers performing rituals at the dam (Z3). Participants in all communities reported risks from animals (e.g., snakes (Z1, Z3, Z6), hyenas or jackals (Z2, Z6), baboons (Z5)). |
| **Physical effects of water collection** | In 2 communities (G1,G3), women reported discomfort and pain in in various parts of the body from water collection. A participant in commu- nity G4, reported that she has a hard time carrying heavy things, including water, since a caesarian section. A pregnant participant in community G6 reported that she had to reduce the amount of water she carried and now needs help from her husband. No data were reported from the remaining 3 communities. | Participants across communities reported fatigue, exhaustion, and physical discomfort of some type related to water collection. In one community (H4), a participant reported having fallen while going to the source, and in another (H6), a participant had sprained an ankle. | Women in all communities mentioned exhaustion and pain in the body. In one community (K2), a participant reported possible miscarriages because of water collection. Women in community K6 reported having more energy because they accessed water from the new bore- hole, which made their water collection easier. | Across communities, women reported numbness, exhaustion, and pain in the neck, back, chest, and other body parts. A participant in community Z1 noted that the journey is harder on her physically when she uses a wheelbarrow because it is heavy and hard to push. In commu- nities with boreholes, one participant (Z3) noted that pumping at the borehole requires a lot of effort and energy and another reported shoulder pain from the pumping (Z6). |

*(Continued)*

Table 2. (Continued)

| Country & Theme | Guatemala (7 communities, 22 Participants) | Honduras (6 communities, 17 Participants) | Kenya (6 communities, 21 Participants) | Zimbabwe (6 communities, 33 Participants) |
|---|---|---|---|---|
| **Children** | Across all communities, some participants carried children when collecting water and some reported that older children may assist with water collection. | Across all communities, women reported being accompanied by children. In 2 communities (H1, H2), participants were observed with children. In one community (H2), one participant carried a toddler when she did not want to walk, and another had two children with her (the older child carried the younger child). Participants in 3 communities (H1, H2, H3) reported that children may help with carrying if they are old enough. | Women across all communities reported that children occasionally collected water to help their mothers but are not the primary water collectors. Children are often left at home to be watched by other women (K4, K5, K6) or older siblings (K2, K3) while the mother collects water. In only one community (K5) was a participant observed to carry a child on a water journey. In 3 communities (K2, K4, K5), mothers with small children reported that they sometimes carry them while collecting water. One woman (community 2) said they carry water on their backs and babies in front. | In 4 of 6 communities (Z1, Z2, Z5, Z6), at least one participant was observed carrying a child while collecting water; women in all communities said they sometimes carried a child while collecting water. In 5 communities (Z1–Z4, Z6), older children were reported to help with water collection or occasionally collected by themselves. Children under 18 were reported to not be allowed to collect water at the river (community Z3). One pregnant participant (community Z5) with a small child had someone mind the child while she collected water. |
| **Other work done during water collection** | Women across all communities reported washing clothing at water points. For many participants, trips to water points for washing clothes were said to be separate from trips for water collection. Women from 4 communities (G2, G4, G5, G6) said they would wash utensils or dishes at sources as well; other women carried water home to wash dishes or utensils there. | Participants across all communities reported washing clothing at the water point. Participants in 3 communities (H1, H2, H4) also reported collecting firewood on their way to or from water points. Participants in 5 communities (H1-H5) commonly bathed themselves or children at water sources. In one community (H6), a participant also reported going to the grocery store on her way to the water point. | Women across all communities reported collecting firewood on the way to or from water sources and bathing children and washing clothes while at water collection points. Women in 4 communities (K2, K3, K4, K5) also took goats to get water at collection sites. | Women reported participating in various activities when collecting water. In all communities, women reported collecting firewood on the journey (except community Z6) and washed clothes at the water source. In 4 communities (Z1, Z2, Z3, Z6), participants reported picking fruit when in season. Some women also watered gardens by the water source (Z5, Z6) and one woman said she harvested mopane worms (Z2). |
| **Perceived water quality and water treatment** | Perceptions of water quality varied across communities and by sources. Women in some communities discussed treating water by boiling (G2, G4) or filtering (G4, G5). Participants in all communities reported work to maintain water sources to preserve the cleanliness of the water (e.g., cleaning the inside of wells (G1, G2, G3) or tanks (G6, G7), or cleaning mud and debris out of areas where spring or surface water is collected (G4, G5, G6, G7). | Perceptions of water quality varied widely between communities and by sources. Women in five communities (H1, H3, H4, H5, H6) said that river or stream water was contaminated. In three communities (H1, H2, H4), women reported that the river gets dirty in the rain. One woman (H3) perceived water from her primary well to be safe for consumption. Participants were not asked about treating water. | Perceptions of water quality varied by community and source. Women in 3 communities (K2, K3, K4) perceived dam water to be the most contaminated because the dams are used by animals. Women in all communities poured out the initial water from dug wells and then collected clearer water eventually seeping through. Women in 5 of 6 communities (K1, K2, K4, K5, K6) reported that they had at some point treated water with chlorine but do not regularly or currently. A woman in community 3 said she would boil water if children had diarrhea. | Across communities, perceptions and practices related to water quality varied. Dam water was perceived to be unsafe because animals drink the water (Z1, Z3). In 3 communities (Z2, Z3, Z6), participants reported treating water (sieves, tablets, or boiling) though some said boiling and tablets negatively affected the taste of water [6]. Water safety techniques (e.g., putting lids on buckets, using one utensil to extract drinking water, separating drinking from other water) were reported (Z5, Z6). Women typically considered the borehole water to be clean, though one woman (Z5) reported borehole water to be rusty. Neither the water quality nor treatment activities were mentioned in community Z4. |

*Community names are not shared. We assigned ID numbers for readers to see where information is from across themes.

**Table 3. Water journey activity, source, time, distance, caloric expenditure, and elevation ascent, by country.**

| | Country | | | | | | | | | |
|---|---|---|---|---|---|---|---|---|---|---|
| | Guatemala (n=22; 7 communities) | | Honduras (n=17; 6 communities) | | Kenya (n=22; 6 communities) | | Zimbabwe (n=33; 6 communities) | | Total (n=94; 25 communities) | |
| Number of water journeys[1] | 23 | | 17 | | 22 | | 33 | | 95 | |
| Primary water journey activity[1] (n, %) | | | | | | | | | | |
| Water collection | 12 | 52.2 | 17 | 100.0 | 19 | 86.36 | 33 | 100.0 | 81 | 85.26 |
| Washing clothes/ laundry | 9 | 39.1 | 0 | 0.0 | 1 | 4.55 | 0 | 0.0 | 10 | 10.53 |
| Water collection & washing clothes | 2 | 8.7 | 0 | 0.0 | 2 | 9.09 | 0 | 0.0 | 4 | 4.21 |
| Time in minutes[1–3] | | | | | | | | | | |
| n | 22 | | 16 | | 22 | | 33 | | 93 | |
| Mean, SD | 67.7 | 52.1 | 38.1 | 19.9 | 164.4 | 87.3 | 57.7 | 45.7 | 81.9 | 73.3 |
| Median | 58.5 | | 31.7 | | 189.3 | | 47.6 | | 53.4 | |
| Range (min, max) | (13.3 - 194.5) | | (18.3 - 99.5) | | (19.2 - 287.2) | | (10.8 - 282.0) | | (10.8 - 287.2) | |
| Difference (shortest & longest) | 181.2 | | 81.2 | | 268.0 | | 271.2 | | 276.4 | |
| Distance (km)[1–3] | | | | | | | | | | |
| n | 22 | | 16 | | 22 | | 33 | | 93 | |
| Mean, SD | 1.1 | 0.8 | 1.1 | 0.7 | 8.5 | 4.8 | 2.8 | 1.8 | 3.5 | 3.9 |
| Median | 0.9 | | | 0.9 | | 9.0 | | 2.4 | 1.9 | |
| Range (min, max) | (0.2 - 3.7) | | (0.2 - 2.7) | | (0.7 - 15.8) | | (0.2 - 8.7) | | (0.2 - 15.8) | |
| Difference (shortest & longest) | | 3.5 | | 2.5 | | 15.1 | | 8.5 | | 15.6 |
| Caloric expenditure (kcal)[1–3] | | | | | | | | | | |
| n | 22 | | 16 | | 22 | | 33 | | 93 | |
| Mean, SD | 117.9 | 87.6 | 143.9 | 94.2 | 421.3 | 218.8 | 223 | 171.6 | 231.4 | 193.5 |
| Median | 97.5 | | | 103.5 | | 489.5 | | 175.0 | 161.0 | |
| Range (min, max) | (36 - 393) | | (42 - 354) | | (65 - 736) | | (37 - 952) | | (36 - 952) | |
| Difference (least & greatest) | | 357 | | 312 | | 671 | | 915 | | 916 |
| Total elevation ascent (m)[1,2,4] | | | | | | | | | | |
| n | 22 | | 15 | | 22 | | 33 | | 92 | |
| Mean, SD | 57.0 | 61.7 | 44.0 | 18.8 | 89.2 | 56.9 | 40.0 | 34.4 | 56.5 | 49.8 |
| Median | 35.5 | | 45.0 | | 86.4 | | 36.0 | | 42.0 | |
| Range (min, max) | (6.1 - 299.0) | | (12.0 - 74.0) | | (7.0 - 214.9) | | (1.0 - 200.0) | | (1.0 - 299.0) | |
| Difference (least & greatest) | | 292.9 | | 62.0 | | 207.9 | | 199.0 | | 298.0 |

[1] 1 participant completed two separate water journeys to different water sources (Guatemala)

[2] 1 participant was excluded from aggregate analyses for time, distance, caloric expenditure, and ascent because she took a bus for part of her water journey and was only on foot for part of the time. Exclusion ensures that the values noted here reflect journeys taken completely on foot. Her specific journey lasted 32.6 minutes, covered 5.4 km, burned 103 calories, and resulted in a gain of 61.3 meters. The distance and ascent values are greater than 1 standard deviation higher than the country mean for these variables. The weight of what she carried is included in all analyses as she did have to carry items to and from the source, even if on the bus part of the way. The total weight of water she collected was 11.3 kg (11.3 L) (Guatemala)

[3] 1 missing & 1 one-way observation was doubled to estimate roundtrip metrics (Honduras)

[4] 2 missing (Honduras)

observed collecting firewood (Honduras, Zimbabwe, Kenya), going to the store (Honduras), harvesting mopane worms (Zimbabwe), and watering goats (Kenya), all of which can lengthen the journey time.

**Methods of carriage.** Most women transported filled water containers without assistance (carried or rolled: 97%), while a few used assistance (wheelbarrow or vehicle: 3%). Women in Honduras, Guatemala, and Zimbabwe carried water on their heads and women in Kenya carried on their back, often assisted by a rope on their head, or by dragging or rolling

the water container behind them. Carrying strategies depended on the size of the vessel and what else was being carried, like children. In Kenya, women said they once had donkeys to help carry water, but many donkeys had died due to the extended drought. One woman followed in Guatemala paid to travel by bus part of the way. Some in Zimbabwe used carts or wheelbarrows to collect larger quantities. One Zimbabwean woman described the work of using wheelbarrows: "When I am using a wheelbarrow, I make sure that I rest at least two times on my way back home from the dam, because it is very hard to push a wheelbarrow along the way. Plus, I will be carrying three buckets of water that are heavy. When carrying on my head, I rest for about two to three times on my way back."

**Frequency of collection.** Women may go to different sources throughout the course of the day or week and the season. Factors such as the weather, family needs, the volume they can carry, other responsibilities (e.g., childcare), available support for carrying water (e.g., other people, wheelbarrow), queues at the water source, and distance to sources influence frequency and source selected. In Honduras and Guatemala, the frequency ranges were the widest. In Honduras, one participant said she collected water once every two weeks, economizing the water to make it last; another said she would go at least ten times per day in the summer. In Guatemala, one participant collected water once per day because she paid for transport to carry the heavy loads; another said she would go up to 20 times in a day if she needed to collect water for bathing. In Zimbabwe, most women reported going to (often varied) water sources several times a day, though the number of trips could depend on their access to a scotch cart (a wooden cart typically pulled by donkeys) or wheelbarrow to transport greater volumes of water. In Kenya, women typically collected water only once per day because of the heat and the distance to water sources. Some women in Kenya would go more than once, but not to riverbeds in the dry season. Those with borehole access would sometimes go twice, but long lines and limited operating hours were barriers.

## 3.3 The burden of collection

The time and distance traveled, the elevation gained, the terrain and risks encountered, the energy expended, the weights carried, and the volumes of water collected varied within and across countries.

**Time.** Across countries, the mean water journey time—including going to the source, completing activities at the source (e.g., extracting water, washing clothes), and returning was 82min (SD = 73.3) (Table 3). Mean times were shortest in Honduras (38min), comparable in Zimbabwe (58min) and Guatemala (68min), and markedly greater in Kenya (164min). Within each country, the shortest journeys were less than 20 minutes (Guatemala: 13min; Honduras: 18min; Kenya: 19min; Zimbabwe: 11min). In three countries, the longest journeys lasted over three hours, and over 4.5 hours in two (Guatemala: 195min; Honduras: 100min; Kenya: 287min; Zimbabwe: 282min).

**Distance.** The mean round-trip distance for all water journeys was 3.5km (SD = 3.9) (Table 3). Mean distances were shortest in Guatemala (1.1km) and Honduras (1.1km), greater in Zimbabwe (2.8km), and greatest in Kenya (8.5km). Within each country, the shortest journeys were all less than 1km (Guatemala: 0.2km; Honduras: 0.2km; Kenya: 0.7km; Zimbabwe: 0.2km) and the farthest were all over 2km, with the longest in Kenya (Guatemala: 3.7km; Honduras: 2.7km; Kenya: 15.8km; Zimbabwe: 8.7km). As Fig 2 shows, those walking the greatest distances and taking the greatest amount of time to collect water were largely doing so to get water from unimproved sources.

**Total elevation ascent.** The overall mean total elevation ascent across countries was 57m (SD = 49.8), with the smallest mean values in Zimbabwe (40m) and Honduras (44m), larger values in Guatemala (57m), and the largest in Kenya (89m) (Table 3). For each country, the smallest total elevation ascent values were all under 15m (Guatemala: 6m; Honduras: 12m; Kenya: 7m; Zimbabwe: 1m). Three of the largest total elevation ascent values were at or over 200m (Guatemala: 299m; Honduras: 74m; Kenya: 214m; Zimbabwe: 200m).

**Environmental risks.** Women traversed challenging terrain and encountered risks while going to water points. Paths were particularly challenging in Guatemala and Zimbabwe. Women in Guatemala went down steep and fragile hillsides. In Zimbabwe, women described hilly and mountainous terrain with loose stones and portions with sand that could burn feet

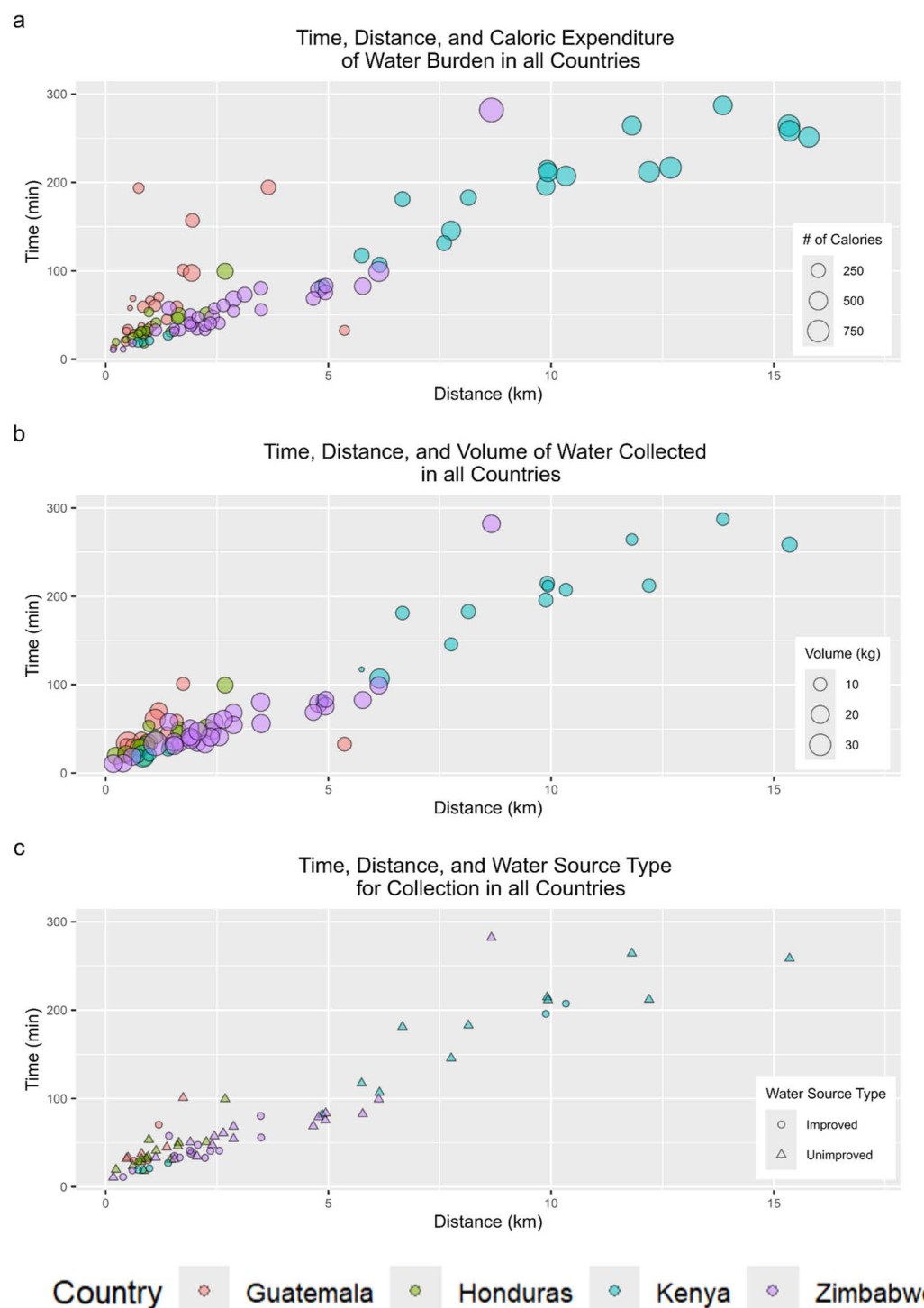

**Fig 2. Time, distance, and (a) caloric expenditure, (b) volume of water collected, and (c) source type among participants followed in Guatemala, Honduras, Kenya, and Zimbabwe.** (Note: The graphs include data from a woman from Zimbabwe who spent more time, went farther, expended more calories, and collected more water than other women in Zimbabwe. She tended a garden by the source; her work hauling water to it resulted in these increased, outlier values.).

and were hard to walk on or push wheelbarrows through, particularly when carrying heavy loads. The rains made terrain treacherous (Honduras) or even more challenging to traverse (Guatemala) because of overflowing rivers and slippery slopes. Women in Guatemala, Kenya, and Zimbabwe said other people could pose risks. In Zimbabwe, women were concerned about fast motorbikes and experiencing sexual harassment and assault. Women reported animals to be risks in Guatemala (snakes, dogs), Honduras (snakes, coyotes), Kenya (elephants, leopards) and Zimbabwe (bees, snakes, hyenas, jackals, and baboons). Women navigated challenging terrain and animal encounters while carrying cumbersome loads:

*"There are many stones, big and small, along the way, making it difficult to walk even when carrying an empty bucket. I once fell and broke my bucket that was full of water. The path is also infested with snakes as they come to seek shelter from the thorny trees."* (Zimbabwe)

**Caloric expenditure.** The overall mean caloric expenditure was 231kcal (SD = 193.5) (See Table 3). Mean expenditure was the least in Guatemala (118 kcal) and Honduras (144 kcal), greater in Zimbabwe (223 kcal) and greatest in Kenya (421 kcal). Within each country, the smallest caloric expenditure values were all under 100 kcal (Guatemala: 36 kcal; Honduras: 42 kcal; Kenya: 65 kcal; Zimbabwe: 37 kcal) and the greatest were all over 350 kcal, (Guatemala: 393 kcal; Honduras: 354 kcal; Kenya: 736 kcal; Zimbabwe: 952 kcal). There is a trend between the water journey time, distance, and calories expended. Most participants from Kenya spent more time, traveled farther, and expended more calories than all other participants (Fig 2).

**Weight of loads carried and volumes of water collected.** The weight and volumes of water women carried varied depending on why they were going to the source (e.g., water collection, laundry), if they were carrying children, and whether they were traveling to or returning from sources (Table 4).

The mean weights of what was carried to water points were higher in Guatemala (6.2 kg) and Zimbabwe (3.4 kg) compared to Honduras (1.4 kg) and Kenya (2.0 kg). Empty water containers, which were carried by over half of women followed in Guatemala (12/22) and by most women followed in Honduras, Kenya, and Zimbabwe weighed less than 1 kg on average (range: 0.2 kg in Honduras to 3.3 kg in Kenya). Children were carried in all countries, contributing to the loads carried to and from the sources. More women were observed carrying children in Zimbabwe [7] and Guatemala [6] than in Honduras [1] and Kenya [1]. Among these women, the mean child weight was 9.3 kg (range: 5.6-12.5 kg, both in Guatemala). Dry laundry was carried to water sources by 13 women, all from Guatemala [11] and Kenya [2]. The two in Kenya carried less than 2kgs of dry laundry weight. In Guatemala, the mean dry laundry weight carried was 6.5kgs (range: 2.5-14.3 kg). Sixteen women (from Guatemala, Kenya, and Zimbabwe) carried other items [range: 0.2 kg for a rope (Kenya) or item to scoop water from dug wells (Zimbabwe) to 11 kg for a bucket of manure carried one-way for gardening (Zimbabwe)].

The mean total weight of loads carried or transported from water sources was 19.3 kg. The greatest loads were observed in Guatemala (43.7 kg) and Zimbabwe (50.1 kg) compared to Honduras (31.7 kg) and Kenya (33.6 kg). The mean proportional increase in weight carried from the source compared to the journey to the source was 1512%. The increase in weights carried for the return journey is attributable to water, whether women brought back water for later use or wet laundry. The mean weight of containers filled with water during return trips was 17.0 kg. The smallest weight of a filled container was observed in Honduras (5.0 kg) and the greatest was observed in Zimbabwe (40.1 kg). Women in Guatemala who brought wet laundry back from the source carried a mean weight of 18.3 kg (range: 8.8-36.5 kg), representing a mean increase 11.8 kg over the dry laundry (range: 5.6-26.6 kg).

The overall mean volume of water collected was 16.1L. The smallest volume of water collected was observed in Kenya (3.7L) and the most was in Zimbabwe (38.2L). Across countries, the mean volumes of water collected and carried were greatest in Zimbabwe (19.6L) followed by Guatemala (15.3L), Honduras (14.8L) and Kenya (12.0L). One women from

PLOS Global Public Health

**Table 4. Water journey transport method and item weight, by country.**

| | Country | | | | | | | | | |
|---|---|---|---|---|---|---|---|---|---|---|
| | Guatemala[1] (n = 23) | | Honduras (n = 17) | | Kenya (n = 22) | | Zimbabwe (n = 33) | | Total (n = 95) | |
| **Water container transport method[2] (n, %)** | | | | | | | | | | |
| Carried (arms, head, back with rope) or rolled | 22 | 95.7 | 16 | 94.1 | 22 | 100.0 | 30 | 96.8 | 90 | 96.8 |
| Pushed wheelbarrow | 0 | 0.0 | 1 | 5.9 | 0 | 0.0 | 1 | 3.2 | 2 | 2.2 |
| Used vehicle | 1 | 4.4 | 0 | 0.0 | 0 | 0.0 | 0 | 0.0 | 1 | 1.1 |
| **Weight carried (kg)** | | | | | | | | | | |
| *Total Weight (kg) Carried to Water Point3* | | | | | | | | | | |
| n | | 23 | | 17 | | 19 | | 33 | | 92 |
| Mean, SD | 6.2 | 5.2 | 1.4 | 3.1 | 2.0 | 1.8 | 3.4 | 5.2 | 3.4 | 4.6 |
| Median | | 5.2 | | 0.6 | | 1.8 | | 0.9 | | 1.0 |
| Range (min, max) | (0.4-17.1) | | (0.2-13.4) | | (0.4-8.4) | | (0.7-22.8) | | (0.2-22.8) | |
| *Total Weight (kg) Carried from Water Point4* | | | | | | | | | | |
| n | | 21 | | 17 | | 18 | | 31 | | 87 |
| Mean, SD | 21.6 | 10.5 | 16.2 | 6.0 | 14.0 | 6.4 | 22.5 | 6.8 | 19.3 | 8.3 |
| Median | | 20.8 | | 16 | | 11.6 | | 20.6 | | 19.4 |
| Range (min, max) | (8.8-43.7) | | (5.0-31.7) | | (6.1-33.6) | | (14.9-50.1) | | (5.0-50.1) | |
| *Total Weight (kg) Increase on Return Journey5* | | | | | | | | | | |
| n | | 21 | | 17 | | 17 | | 31 | | 86 |
| Mean kg increase, SD | 14.9 | 8.6 | 14.8 | 4.5 | 12.1 | 6.4 | 19.4 | 4.0 | 15.9 | 6.5 |
| Median | | 11.3 | | 15.5 | | 10.5 | | 19.2 | | 16.5 |
| Range - kg (min, max) | (5.6-35.3) | | (4.6-21.1) | | (3.7-31.3) | | (13.4-38.2) | | (3.7-38.2) | |
| Mean % increase compared to weight carried to point | | 706% | | 2574% | | 1004% | | 1740% | | 1512% |
| Range - % (min, max) | (41%-2475%) | | (137%-9150%) | | (125%-3025%) | | (59%-2886%) | | (41%-9150%) | |
| *Weight (kg) of Water Containers, Empty6* | | | | | | | | | | |
| n | | 12 | | 17 | | 19 | | 33 | | 81 |
| Mean, SD | 0.7 | 0.4 | 0.7 | 0.3 | 1.4 | 0.8 | 0.9 | 0.2 | 0.9 | 0.5 |
| Median | | 0.7 | | 0.6 | | 1.1 | | 0.9 | | 0.8 |
| Range (min, max) | (0.4-2.0) | | (0.2-1.5) | | (0.4-3.3) | | (0.7-1.9) | | (0.2-3.3) | |
| *Weight (kg) of Water Containers, With Water7* | | | | | | | | | | |
| n | | 12 | | 17 | | 18 | | 31 | | 78 |
| Mean, SD | 15.9 | 8.0 | 15.4 | 4.6 | 13.4 | 6.4 | 20.5 | 4.0 | 17.0 | 6.1 |
| Median | | 13.2 | | 16.0 | | 11.6 | | 20.1 | | 18.3 |
| Range (min, max) | (8.8-36.0) | | (5.0-21.7) | | (6.1-33.6) | | (14.9-40.1) | | (5.0-40.1) | |
| *Weight (kg) and Volume (L) of Water (1 kg water = 1L)8* | | | | | | | | | | |
| n | | 12 | | 17 | | 18 | | 31 | | 78 |
| Mean, SD | 15.3 | 7.9 | 14.8 | 4.5 | 12.0 | 6.2 | 19.6 | 3.8 | 16.1 | 6.1 |
| Median | | 12.9 | | 15.5 | | 10.5 | | 19.3 | | 17.2 |
| Range (min, max) | (8.4-35.3) | | (4.6-21.1) | | (3.7-31.1) | | (14.0-38.2) | | (3.7-38.2) | |
| *Calories (kcal) Expended per Liter of Water Carried9* | | | | | | | | | | |
| n | | 12 | | 16 | | 18 | | 31 | | 77 |
| Mean, SD | 7.8 | 4.5 | 11.1 | 7.9 | 39.3 | 24.5 | 11.8 | 9.4 | 17.4 | 18.3 |
| Median | | 6.2 | | 8.9 | | 41.0 | | 8.9 | | 9.4 |
| Range (min, max) | (2.9-16.6) | | (2.7-27.4) | | (2.1-79.5) | | (1.9-49.1) | | (1.9-79.5) | |

*(Continued)*

| | Country | | | | | | | | | |
|---|---|---|---|---|---|---|---|---|---|---|
| | Guatemala[1] (n = 23) | | Honduras (n = 17) | | Kenya (n = 22) | | Zimbabwe (n = 33) | | Total (n = 95) | |
| **Weight (kg) of Carried Children** | | | | | | | | | | |
| n | | 6 | | 1 | | 1 | | 7 | | 15 |
| Mean, SD | 8.5 | 2.4 | 12.8 | NA | 7.1 | NA | 9.7 | 1.8 | 9.3 | 2.3 |
| Median | | 8.2 | | 12.8 | | 7.1 | | 10 | | 9.8 |
| Range (min, max) | (5.6-12.5) | | NA | | NA | | (6.7-12.2) | | (5.6-12.5) | |
| **Weight (kg) of Other Items Carried10** | | | | | | | | | | |
| n | | 6 | | 0 | | 2 | | 8 | | 16 |
| Mean, SD | 1.8 | 1.7 | | | 0.7 | 0.6 | 1.9 | 3.8 | 1.7 | 2.8 |
| Median | | 1.1 | | | | 0.7 | | 0.2 | | 0.7 |
| Range (min, max) | (0.6-5.2) | | | | (0.2-1.1) | | (0.2-11.0) | | (0.2-11.0) | |
| **Weight (kg) of Dry Laundry11** | | | | | | | | | | |
| n | | 11 | | 0 | | 2 | | 0 | | 13 |
| Mean, SD | 6.5 | 3.6 | | | 1.8 | 0.1 | | | 5.8 | 3.7 |
| Median | | 4.7 | | | | 1.8 | | | | 4.5 |
| Range (min, max) | (2.5-14.3) | | | | (1.7-1.8) | | | | (1.7-14.3) | |
| **Weight (kg) of Wet Laundry12** | | | | | | | | | | |
| n | | 11 | | 0 | | 2 | | 0 | | 13 |
| Mean, SD | 18.3 | 9.0 | | | 1.9 | 0.8 | | | 15.8 | 10.3 |
| Median | | 15.6 | | | | 1.9 | | | | 14.3 |
| Range (min, max) | (8.8-36.5) | | | | (1.3-2.5) | | | | (1.3-36.5) | |
| **Laundry Weight Increase (Dry to Wet)13** | | | | | | | | | | |
| n | | 11 | | 0 | | 1 | | 0 | | 12 |
| Mean kg increase, SD | 11.8 | 6.3 | | | 0.7 | NA | | | 10.9 | 6.8 |
| Median | | 10.9 | | | | 0.7 | | | | 10.0 |
| Range kg increase (min, max) | (5.6-26.6) | | | | NA | | | | (0.7-26.6) | |
| Mean % increase | 200% | | | | NA | | | | 187% | |
| Range % increase (min, max) | (96%-362%) | | | | NA | | | | (39%-363%) | |
| **Participant Body Weight (kg)** | | | | | | | | | | |
| n | | 22 | | 17 | | 22 | | 33 | | 94 |
| Mean, SD | 56.9 | 8.5 | 65.4 | 14.5 | 48.2 | 7.9 | 64.4 | 12.1 | 59.0 | 12.8 |
| Median | | 56.8 | | 65.0 | | 49.7 | | 62.7 | | 56.8 |
| Range (min, max) | (43.8-84.0) | | (44.4-94.6) | | (35.8-64.2) | | (48.9-98.0) | | (35.8-98.0) | |
| **Weight (kg) of Containers with Water as Proportion of Participant Body Weight14** | | | | | | | | | | |
| n | | 12 | | 17 | | 18 | | 31 | | 78 |
| Mean | | 29% | | 23% | | 25% | | 32% | | 28% |
| Range (min, max) | | (15%-72%) | (6%-39%) | | (9%-61%) | | (16%-63%) | | (6%-72%) | |

[1] 1 participant completed two separate water journeys to different water sources (Guatemala)

[2] 2 missing (Zimbabwe)

[3] 3 missing (Kenya)

[4] 2 missing (Guatemala); 4 missing (Kenya); 2 missing (Zimbabwe)

[5] 2 missing (Guatemala); 5 missing (Kenya); 2 missing (Zimbabwe)

[6] 11 missing (Guatemala); 3 missing (Kenya)

[7] 11 missing (Guatemala); 4 missing (Kenya); 2 missing (Zimbabwe)

*(Continued)*

**Table 4.** (Continued)

[8]11 missing & 2 include weight of water container in calculation (Guatemala); 4 missing & 1 includes weight of water container in calculation (Kenya); 2 missing (Zimbabwe)

[9]11 missing (Guatemala); 1 missing (Honduras); 4 missing (Kenya); 2 missing (Zimbabwe)

[10]*Other items:* soap, basins/washbowls, dishes, brooms (Guatemala); rope (Kenya); scooping plates, manure, vegetables (Zimbabwe)

[11]1 missing (Kenya)

[12]1 missing (Kenya)

[13]2 missing (Kenya)

[14]10 missing & 1 participant was excluded from aggregate analyses for time, distance, caloric expenditure, and ascent because she took a bus for part of her water journey and was only on foot for part of the time. The weight of what she carried is included in all analyses as she did have to carry items to and from the source, even if on the bus part of the way (Guatemala); 4 missing (Kenya); 2 missing (Zimbabwe)

Zimbabwe reflected on the limited quantity of water she collects and how she manages: "[If] we had a lot of water, we would be using water for many different reasons; but due to unavailability of water we are minimizing the water so that it can accommodate all of the activities that we are undertaking. Therefore, we normally get water from the far sources; and yet we are able to carry a small jerrican." (Zimbabwe)

On average, women expended 17.4 calories per liter collected; the mean number of calories expended per liter of water carried were comparable in Guatemala (7.8kcal/L), Honduras (11.1kcal/L), and Zimbabwe (11.8kcal/L), and were much greater in Kenya (39.3kcal/L). The least amount of calories exerted for one liter of water collected was in Zimbabwe (1.9kcal/L), and the greatest was in Kenya (79.5kcal/L).

The water weight that women carried or transported was equivalent to substantial proportions of their body weight. The mean proportion of collected water weight to participant body weight was 28%. Across countries, values were comparable (Guatemala: 29%; Honduras: 23%; Kenya: 25%; Zimbabwe: 32%). The smallest proportion was recorded in Kenya (9%) and the largest proportion in Guatemala (72%). (Note, the Guatemalan woman bearing this weight was carrying it unassisted (see Table 4)). As Fig 2 shows, there is a trend between the journey time, distance, and volume (or weight) carried. Specifically, women in Kenya, who were experiencing a drought, walked the greatest distances, took the greatest time fetching water and collected smaller volumes of water. Conversely, higher volumes of water were collected by those traveling shorter distances for less time.

**Physical and mental impacts.** Women reported myriad physical impacts of water collection from carrying heavy loads of water, traversing challenging terrain, and expending effort going to and working at sources. Women discussed general discomfort, pain (neck, back, chest, shoulders, arms, knees, legs, stomach, ribs), headache, shortness of breath, blisters, weakness, numbness, fatigue, poor sleep, and exhaustion. As one woman in Kenya commented, "You feel a headache after you finish all the tasks; you feel very tired. You cannot even sleep well due to the tiredness. You feel pain in your ribs, on your back, headache; so you feel pain all over your body."

Women reported specific injuries related to animal encounters or falling, whether experienced by them or those they know. Pregnancy was said to exacerbate discomfort and pain (Kenya) and require women to carry smaller quantities of water (Guatemala). Women in all countries reported falling and slipping, and those in Guatemala specifically reported sprained ankles and broken legs because of the mountainous terrain, made riskier if barefoot. As one woman shared:

*"Once, my daughter broke her leg while fetching water here. She didn't like going to get water, so she carried a larger jar on her side. During one of these trips, she slipped, and her foot got trapped, leading to the accident. I often advised her to make multiple trips to ease the burden, but she preferred carrying it all at once."* (Guatemala)

Finally, women commented on the mental burden of water collection, the need to ration, and the implications of not having water, which can be stressful. One woman from Honduras noted the deliberation involved in water use given the effort to get water:

*"In fact, I ration its use because it really represents an effort for me. I need to evaluate when it will be possible to use it, since bringing it involves a long journey. Therefore, I use it in a controlled manner, treating it with great care due to the effort involved in obtaining it. Furthermore, I am very clear for what purposes I will use it. I don't waste it unnecessarily because I know that getting it is difficult."* (Honduras)

### 3.4 Alignment of estimated and measured water collection time

On average, those who indicated they would use the water for drinking (n = 30) over-estimated by 39 minutes and those who collected for 'other uses' (n = 6) under-estimated by 11 minutes (Table 5). Average estimates from participants in Honduras, Zimbabwe, and Guatemala were not far from the average measured times, and times were greatly over-estimated by Kenyan participants. Specifically, participants underestimated across all water collection activities by 19 minutes in Guatemala and by 2 minutes in Honduras and overestimated by 11 minutes in Zimbabwe. In Kenya, participants overestimated by an average of 85 minutes. About half (47%) of participants estimated their water collection time to be within 15 minutes of the measure time (S2 Table). Among those whose estimates were greater than 15 minutes off, a greater proportion overestimated (42%) than under-estimated (11%).

**Table 5. Estimated and measured times to collect water overall (n = 36), stratified by if for drinking (n = 30) or other uses (n = 6).**

| | Estimated time (min) | | | | Measured time (min) | | | | Difference (estimated - measured min) | | |
|---|---|---|---|---|---|---|---|---|---|---|---|
| | n | Mean | Range | SD | Mean | Range | SD | Mean | Range | SD |
| **All water collection[1]** | | | | | | | | | | |
| All | 36 | 115.4 | (10.0 - 520.0) | 141.2 | 85.0 | (18.3 - 264.4) | 77.7 | 30.4 | (-121.1 - 303.3) | 81.6 |
| Guatemala | 4 | 18.8 | (10.0 - 30.0) | 8.5 | 38.2 | (20.4 - 70.40) | 22.1 | -19.4 | (-60.4 - (-0.1)) | 27.7 |
| Honduras | 5 | 28.0 | (20.0 - 30.0) | 4.5 | 29.6 | (18.3 - 41.1) | 8.1 | -1.6 | (-11.1 - 11.7) | 9.1 |
| Kenya | 12 | 258.3 | (60.0 - 520.0) | 169.4 | 173.1 | (20.9 - 264.4) | 76.6 | 85.3 | (-121.1 - 303.2) | 123.0 |
| Zimbabwe | 15 | 56.0 | (20.0 - 105.0) | 24.7 | 45.4 | (18.7 - 83.4) | 17.8 | 10.6 | (-20.3 - 65.5) | 22.7 |
| **Drinking water** | | | | | | | | | | |
| All | 30 | 132.5 | (15.0 - 520.0) | 149.0 | 93.8 | (18.7 - 264.4) | 82.1 | 38.7 | (-121.1 - 303.2) | 86.7 |
| Guatemala | 3 | 21.7 | (15.0 - 30.0) | 7.6 | 27.4 | (20.4 - 31.8) | 6.2 | -5.7 | (-11.8 - (-0.1)) | 5.9 |
| Honduras | 3 | 26.7 | (20.0 - 30.0) | 5.8 | 33.8 | (28.8 - 41.1) | 6.4 | -7.1 | (-11.1 - (-1.6)) | 5.0 |
| Kenya | 12 | 258.3 | (60.0 - 520.0) | 169.4 | 173.1 | (20.9 - 264.4) | 76.6 | 85.3 | (-121.1 - 303.2) | 123.0 |
| Zimbabwe | 12 | 60.8 | (20.0 - 105.0) | 23.8 | 46.2 | (18.7 - 83.4) | 18.8 | 14.6 | (-20.3 - 65.5) | 23.6 |
| **Water for other uses[2]** | | | | | | | | | | |
| All | 6 | 30.0 | (10.0 - 60.0) | 16.7 | 40.7 | (18.3 - 70.4) | 20.3 | -10.7 | (-60.4 - 11.7) | 25.6 |
| Guatemala | 1 | 10.0 | (10.0 - 10.0) | | 70.4 | (70.4 - 70.4) | | -60.4 | (-60.4 - (-60.4)) | |
| Honduras | 2 | 30.0 | (30.0 - 30.0) | 0.0 | 23.2 | (18.3 - 28.2) | 7.0 | 6.8 | (1.8 - 11.7) | 7.0 |
| Kenya | 0 | | | | | | | | | |
| Zimbabwe | 3 | 36.7 | (20.0 - 60.0) | 20.8 | 42.4 | (33.1 - 60.9) | 16.0 | -5.7 | (-13.2 - (-0.9)) | 6.6 |

[1]Data from 10 women were excluded because they did not collect water to bring home (9 Guatemala, 1 Kenya) and from 10 women because they did not provide estimated times to collect water (5 Guatemala, 1 Honduras, 4 Zimbabwe). Data from 38 women were excluded because their water source type was not the same for their estimated and measured water journey times (5 Guatemala, 10 Honduras, 9 Kenya, 14 Zimbabwe). For the one Guatemala participant who completed two water journeys, only her water journey to collect water was included

[2]Water was classified as for 'other uses' if the respondent did not explicitly say it would be used for drinking

# 4. Discussion

Our mixed-methods investigation applied novel approaches to comprehensively assess women's lived experiences while traveling to, carrying out work at, and returning from water sources across Guatemala, Honduras, Kenya, and Zimbabwe. Findings illuminate how this unpaid, gendered task demands women's energy, consumes their time, exposes them to risks, and impacts their well-being—often for water from unimproved sources that is of insufficient quantity to ensure needs are met. Our results confirm the hypothesis made by our World Vision SWSW program partners regarding substantial water burden in these areas and support previous calls for water service improvement efforts to prioritize initiatives that reduce the disproportionate time, energy, and physical and mental health burdens of women and ameliorate this gender inequitable burden [19,33].

In our study, the time women spent, the energy they expended, the distances they traveled, and the elevation they gained going to and from water sources diverge widely from the results of previous research, revealing how limited the understanding of women's experiences has been. In the more extreme examples, women in our study spent over 4.5 hours (Kenya and Zimbabwe), burned over 700 calories (Kenya and Zimbabwe), walked almost 16 kilometers (Kenya), and ascended nearly 300 meters in elevation (Guatemala) to access and use water sources. Participants in previous research typically traveled less than 1km with measured times of less than one hour [33,35,41–43] (exceptions: 2018 study in Benin [62]; 2014 study in Mozambique) [45]; previous data tracking caloric expenditure is dated [7] and elevation ascent data could not be found. This study not only provides data that is novel and presents an approach for others to collect similar data in the future, but it also illuminates and expands on our understanding of the burden shouldered for water collection and carrying out work that requires water access. Even if not feasible to mimic the methods of this study in other locations or at scale, those drafting policies and delivering water services can still glean importance insights to inform their work. Specifically, even simple estimates of time, distance, energy, and elevation burden should be considered when assessing the appropriateness of what water services can and should be delivered and who should be prioritized in gaining access to such services.

Journeys that were not as extreme should not be assumed to be of inconsequential impact. Even for women walking shorter distances for less time, our data suggest that this persistent labor amounts to a tremendous inequity of lost time and energy and bodily strain over the course of a lifetime. Moreover, we recognize that impacts are likely greater than what we found. Firstly, we assessed the burden of a single journey, not the burden of a single day. Thus, efforts may be far greater than what we recorded. Secondly, we did not assess financial or opportunity costs. Even those spending less time and energy were still depleting important resources that could have been channeled for other uses. Thirdly, it is possible that all participants felt the effect of a 'bandwidth tax', which is the mental toll or cognitive exhaustion experienced among those living in scarcity due to constant thinking about how to manage limited and tenuous resources [63]. As Ray and Smith (2021) have discussed, insufficient, unreliable, and inadequate water resources can impose such a 'bandwidth tax' [64]. Consistent with other studies [11,12], many of our participants described water journeys as physically and mentally exhausting. Adapting a point by Ray and Smith [64], when people are consumed with managing their most immediate needs—like water needs—they may lack the cognitive energy to plan for or even recognize other current or future needs for themselves or their families. Thus, while our work provides critical and novel data on the water access burden, we expect the burden among all participants is likely greater than what we captured.

Those expending the most energy and time collecting water were getting small quantities and from unsafe sources, raising questions about additional unmeasured impacts on health, particularly for proper hydration, nutrition, and hygiene. We found that on average, women in Kenya fetched once per day as the sole collector in the household, took over 2.5 hours to collect water, expended 40 calories per liter of water, and collected only 12 liters for a household of six. Assuming water is distributed equally, all household members would have just two liters of water per day. Based on the WHO publication on 'Domestic water quantity, service level and health', quantities below 5.3 liters/person/day from sources greater than 1 kilometer away and requiring over 30 minutes of collection time are considered a 'very high' level of health concern

because adequate amounts needed for drinking, cooking, and hygiene cannot be assured [65]. Further, the average adult woman needs at least 2.7L of water per day for drinking alone, and much more with heavy physical activity and in hot, dry environments [66]. Thus, the women collecting water could be spending considerably more energy per liter they are allotted compared to other household members. Further, because data on intrahousehold use are limited and equitable distribution is not certain [9] women could be getting even less water than others, as has been shown with food [67]. Compounding the limited water available and the energy expense, in many countries women eat 'last and least' in the family, potentially leading to an 'energy output-input' imbalance [67]. Thus, women may not be adequately replacing both the water and the calories they expend for water collection. While we did not investigate hydration, diet, body composition (e.g., fat mass), or hygiene, some women in our study did discuss these issues and further work on caloric and water adequacy is warranted.

Consistent with other studies, [11,12,16,63] participants described how water journeys could cause injuries, and our assessment of the amount of weight women carried amplifies specific concerns about musculoskeletal health impacts. A recent scoping review [16] on water carriage suggests that impacts to musculoskeletal health may be substantial, warranting further research. The authors note that musculoskeletal health is not widely assessed, and pain associated with it may be under-reported as women may accept musculoskeletal pain as a consequence of their role as water collectors [16]. Other research argues that, because cervical spinal conditions are a common cause of disability, carrying water, specifically on the head, may be a major contributor to musculoskeletal disease burden in low-income countries [11,68]. Our novel data detailing the extensive weights women carried provide further justification for water infrastructure improvement efforts that center women's health and aspire to eliminate or at least reduce the physical burden of water carriage. Findings also support the need for more research on how water carriage may contribute to musculoskeletal health and disability. However, we emphasize that such research should extend beyond water carriage to encompass all items carried for water-related work given the tremendous non-water weights we observed women to shoulder (e.g., clothing). Based on context, such research could also incorporate firewood collection, a similarly gendered and physically demanding activity [17,69,70].

Our research provides further evidence of multiple water source use and the need for research and monitoring to look beyond collection from primary drinking water sources alone. Water collection from multiple sources has been found to be a global phenomenon—even among those with improved sources [71]—with source selection determined by need, season, and source characteristics, among other factors [18]. In our study, large proportions of women in Guatemala, Honduras, and Zimbabwe reported visiting different sources for drinking and 'other uses.' For some, as we saw in Guatemala, women did not collect water but carried out strenuous laundering activities at the source. The recently published 'Priority Gender-Specific Indicators for WASH Monitoring Under SDG Targets 6.1 and 6.2' recommends new indicators that could shed light on the burden of collecting from multiple sources (i.e., 'Average time primary water collector spends per day collecting drinking water, by sex and age' and 'Average time primary water collector spends per day collecting water for all household needs, by sex and age') [72]. These indicators do not encompass water work at the sources, which, based on our findings, deserves more attention.

Our findings show that self-reported time estimates for water collection are imperfect, but still valuable for monitoring. Half of respondents' estimates were within 15 minutes of the measured time. Most of the participants who over-estimated by more than 15 minutes were from Kenya where water journeys were very long. Field team members remarked that these women often take animals with them to water at the sources, which can lengthen the process substantially. However, they did not bring animals on the days they were followed. If they had, estimates could have been closer to actual water journey times. Those who were closer to sources had more accurate estimations, thus we do not expect that time estimates would interfere with how JMP classifies access for global monitoring, though self-reported time burden assessments may need caution when including those who are particularly far from sources. Further, because we learned about various time-consuming activities taking place at or on the way to the water sources—like watering animals, digging in

river beds to extract water, and laundering clothing—we also suggest caution if assuming that distance to a source can be estimated based on either measured or self-reported time. Our data show that the time involved in water collection can include more than just the journey to and from the source, and that the journey times can vary based on the environments.

Programs that aim to improve water infrastructure should leverage the methodologies we employed to objectively evaluate if programs positively impact the lives of women and girls. Unfortunately, while the myriad benefits of how improved water services can improve the lives of women and girls have been long proclaimed [19,73], rigorous assessments to determine impact are scant. Benefits are not guaranteed and may be (erroneously) assumed, instead of tested. As one study in Zambia demonstrates, water improvements may ease burden for some but shift it to others. Specifically, while researchers found that a new, closer, improved water supply (borehole) significantly decreased the amount of time adult women spent on water-related work, girls who lived close to the new borehole significantly increased the time they spent on water-related work and significantly decreased the amount of time they spent in school and on homework (with no changes observed among boys) [74]. Alternatively, even if closer sources are made available, women may just frequent them more often or shoulder larger loads of water [62]. They may get more water but the reductions in time and energy may not be enough to enable engagement in other activities (e.g., economic) [62]. Or, women may still use farther or unimproved sources if a newer improved one is not always available (as observed in our study among Kenyan women), requires payment (as noted in Benin [62]), or does not meet their specific preferences or needs (as noted in India [18], Ghana [75], and observed in our study among women from Guatemala who carried heavy clothing loads to and from the river for washing despite access to other sources). Thus, evaluations, using approaches like ours, should be used to assess quantitative and qualitative impacts on women's lives.

## 5 Strengths and limitations

Our mixed-methods study provides a comprehensive understanding of women's experiences accessing water sources using novel and state-of-the-art approaches, though limitations exist. Community sampling was purposive, and individual sampling was by convenience, thus data may not be representative, and are not generalizable. The small convenience samples limit our ability to conduct analysis by age or life course strata. For example, while the greatest proportion of participants in Guatemala and Honduras were unmarried with partners and most participants in Kenya and Zimbabwe were married, we are not able to confirm if the water collection responsibilities and experiences we observed are most typical for women of these life stage groups from these countries compared to women of any other life course stages. Further, we did not collect data with any women under age 18 and therefore are missing their experiences. Future work could explore if and how water collection roles, responsibilities, and burdens vary by age and/or life course stages within countries. Such insights could reveal if water fetching burdens are for all women to bear, or if specific groups of women are more likely to shoulder it.

We only assessed a single journey to a water source at a single time of year per enrolled household due to time and resource constraints. Future research could collect data throughout the day and from all water collectors within a household to capture daily burden for individuals and the household more accurately. Further, data reflect the seasonal conditions during which data were collected. Previous research has shown that water fetching burden can be exacerbated during the dry season [24–26]. Data were collected in Zimbabwe and Kenya during the dry season, and thus our data may reflect greater burden compared to what is experienced at other times of the year. Future research could assess variability of water collection and water work burden by collecting data during different seasons.

While those who participated in the study were asked to carry out their water journey as they normally would, we know that our presence may have influenced the source they chose to visit and what they did along the way. For example, some women in Kenya decided to go to unimproved sources despite having a new community-level borehole and others mentioned often taking animals with them, but none did for the journeys we observed and recorded. It is unclear if routines were changed because of participation in data collection, but it is possible. Thus, our data may include journeys that

are different than what they would have been had we not been there, and they could have been either longer or shorter. It is also possible that participants' speeds were different because we were with them. Specifically, they could have moved more slowly because they were answering questions or adjusting to the pace of data collection team members who were not accustomed to the journey. Or, they could have moved faster, feeling pressure to complete the trip or deciding not to linger in conversation with others, bathe, or carry out other activities along the way. Journeys were recorded as a complete experience, preventing assessment of the time and energy burden of various activities within the journey (e.g., the walk to source vs. the walk from source; water extraction).

Caloric expenditure is modeled by watches, which become more precise over time as the watch learns about the individual wearer's behavior. However, watches were worn only once by each participant, so estimates may be imperfect. Transcriptions of some IDIs were not verbatim, but summaries and descriptions of conversations still contributed to the analysis. Finally, we carried out a rapid thematic analysis, which best suited our aim to understand women's practices related to water collection. The qualitative data could be further explored in a subsequent analysis to understand determinants of those practices, if the data enables it.

## 6  Conclusion

While water is necessary to sustain life and thus should be considered a source of nourishment, we show how a lack of adequate and accessible water was actually depleting, draining women of energy and time, and posing risks to their well-being. Crider and Ray (2022) argue that research in the sector often focuses on women for their instrumental value, or how women can support programming, instead of focusing on women's intrinsic worth [9]. Relatedly, MacArthur and colleagues (2023) offer a framework showing that water, sanitation, and hygiene (WASH) programs can either operate with such instrumental objectives or—alternatively—can have gender-transformative objectives, defined as those that "actively seek to transform the gender norms, structures, and dynamics which perpetuate inequalities within households, communities, and institutions" [76]. We agree that programing that seeks to prevent the perpetuation of inequalities is important. We also acknowledge that such approaches may not be possible for those implementing WASH programs to deliver or achieve. As such, we amplify an additional point by MacArthur and colleagues, who note that a more practicable approach may be to both understand existing gender norms and ensure that programs 'do-no-harm' by not leveraging or perpetuating these norms for programmatic gains. Through our use of quantitative and qualitative research methods, we centered women's lived experiences, which have intrinsic value, in hopes of understanding the types of harms to women that could be averted and the potential benefits that women could gain if water programing was more gender aware. The results of our study reinforce the need to redouble efforts to improve water access in low-resource settings, focusing such improvement efforts on providing gender-specific benefits, eliminating harm (even if unintended), and rigorously assessing the impacts of such efforts on women's lives.

## Supporting information

**S1 Checklist. Standards for reporting qualitative research.**
(DOCX)

**S2 Checklist. PLOS global public health inclusivity in global research.**
(DOCX)

**S1 Table. Thematic descriptions of women's water collection experiences and practices, by community.**
(DOCX)

**S2 Table. Water Journey estimated time comparison and discordance, by country.**
(DOCX)

**S1 Text. Protocol for go-along interviews, semi-structured observations, and research equipment.**
(DOCX)

**S2 Text. Data collection tools for participant demographic information, go-along interviews, and semi-structured observations.**
(DOCX)

## Acknowledgments

We are grateful to the participants of this study for their time and for sharing their experiences. We would like to thank the following partners and enumerators for their research collaboration. From Centro Universitario de Occidente, de la Universidad de San Carlos de Guatemala: Raul Bethancourt, Luis Alfredo López Cortez, Leonardo Cabrera López, Facundo Zacarías López Sánchez, Alex Fernando López Cortez, Zaida Anaí Vasquez García, Gladys Judith Vásquez García López, Vilma Rosario López Sánchez, Any Gabriela León Ramon, Heidy Maribel Baten Velásquez, Lucy Antonieta Ixmucané Capriel Herrera, Lesbia Beatriz Xiloj Hernandez, Ana Rosa León, José Joél Nolasco Yacabalquiej, Ingrid Johana Guachiac Gómez; from World Vision Guatemala: Enri Maldonado, Erick Calderón, Estuardo Gonzales; from Universidad Nacional Autónoma de Honduras: Rosaura Suyapa Rodriguez Funez, Ashly Michelle Méndez, Nastin Nahomy Tilguant Ávila, Brenda Paola Cruz, Yesi Karolina Murillo Vega, Jorge Luis Cálix Barahona, Dulce Milagro Sevilla Sosa, Wilber Adonay Ávila Valladares, Néstor Fabricio Dávila González, Pedro Alejandro Hernández; from St. Paul's University, Kenya: Lesoito Jeremiah Christopher, Leshaule Benedicto Alois, Patricia Lekiliyo, Lechalote Ben Ljania, Nasieku Lolchuragi, Lekalantula Raina Deborah, Glory Karimi Kiriinya, Lekonte Salome Mary, Lelikoo Naanyu Sarah, John Kalasinga Lerosion; from World Vision Kenya: Kevin Muche; from Datalyst Africa: Molly Manyonganise, Eve Nyemba, Sunga Mzeche, Nyasha Kamwire, Luckness Zimanyiwa, Sibonisiwe Mpofu, Tapiwa Mapfumo, Kudzai Zibgwi, Pricilla Moyo, Gladys Mhaka, Sithokozile Sibanda, Gugulethu Mnkandhla; and from World Vision Zimbabwe: Nobuhle Mlotshwa. Lastly, we would like to thank Jorge Beteta for his translation services.

## Author contributions

**Conceptualization:** Bethany A Caruso, Sheela S. Sinharoy.

**Data curation:** Thea Mink, Madeleine Patrick, Emily Ogutu, Cameron Dawkins, Olivia Bendit, Mahnoor Fatima, Alicia Macler, Jera White, Alondra Zamora, Gladys Ramos, Peter Koome, Petronilla Andiba Otuya, Paul Ruto.

**Formal analysis:** Bethany A Caruso, Thea Mink, Madeleine Patrick, Cameron Dawkins.

**Funding acquisition:** Bethany A Caruso.

**Investigation:** Gladys Ramos, Peter Koome, Rohin Otieno Onyango, Petronilla Andiba Otuya, Paul Ruto, Munyaradzi Damson.

**Methodology:** Bethany A Caruso, Thea Mink, Madeleine Patrick, Emily Ogutu, Everlyne Atandi, Sheela S. Sinharoy.

**Project administration:** Bethany A Caruso, Thea Mink, Madeleine Patrick, Emily Ogutu, Cameron Dawkins, Olivia Bendit, Mahnoor Fatima, Ingrid Lustig, Alicia Macler, Jera White, Alondra Zamora, Héctor Salvador Peña Ramírez, Carlos Daniel Sic, Sandra Antonio, Everlyne Atandi, Peter Mwangi, Jammaine Jimu, Makaita Maworera, Munyaradzi Damson.

**Resources:** Bethany A Caruso, Héctor Salvador Peña Ramírez, Carlos Daniel Sic, Sandra Antonio, Jazmina Nohemí Irías, Jammaine Jimu, Makaita Maworera.

**Supervision:** Bethany A Caruso, Thea Mink, Madeleine Patrick, Emily Ogutu, Cameron Dawkins, Olivia Bendit, Mahnoor Fatima, Ingrid Lustig, Alicia Macler, Jera White, Alondra Zamora, Carlos Daniel Sic, Sandra Antonio, Jazmina Nohemí Irías, Peter Mwangi, Jammaine Jimu, Makaita Maworera, Munyaradzi Damson, Sheela S. Sinharoy.

**Validation:** Bethany A Caruso, Thea Mink, Madeleine Patrick, Emily Ogutu.

**Visualization:** Bethany A Caruso, Thea Mink, Madeleine Patrick.

**Writing – original draft:** Bethany A Caruso, Thea Mink, Madeleine Patrick.

**Writing – review & editing:** Bethany A Caruso, Thea Mink, Madeleine Patrick, Emily Ogutu, Cameron Dawkins, Olivia Bendit, Mahnoor Fatima, Ingrid Lustig, Alicia Macler, Jera White, Alondra Zamora, Alberto Emanuel Santos López, Héctor Salvador Peña Ramírez, Carlos Daniel Sic, Jorge Lemus Chávez, Sandra Antonio, Jazmina Nohemí Irías, Gladys Ramos, Everlyne Atandi, Peter Mwangi, Peter Koome, Rohin Otieno Onyango, Petronilla Andiba Otuya, Paul Ruto, Morris Chidavaenzi, Jammaine Jimu, Sithandekile Maphosa, Makaita Maworera, Munyaradzi Damson, Sheela S. Sinharoy.

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
