## [Decision Letter · Decision Letter 0]

5 May 2025

PGPH-D-25-00317

Women’s experiences collecting and accessing water in Guatemala, Honduras, Kenya, and Zimbabwe: A mixed-methods investigation

Dear Dr. Caruso,

Thank you for submitting your manuscript to PLOS Global Public Health. After careful consideration, we feel that it has merit but does not fully meet PLOS Global Public Health’s publication criteria as it currently stands. Therefore, we invite you to submit a revised version of the manuscript that addresses the points raised during the review process.

Please note that we have only been able to secure a single reviewer to assess your manuscript. We are issuing a decision on your manuscript at this point to prevent further delays in the evaluation of your manuscript. Please be aware that the editor who handles your revised manuscript might find it necessary to invite additional reviewers to assess this work once the revised manuscript is submitted. However, we will aim to proceed on the basis of this single review if possible. 

We look forward to receiving your revised manuscript.

Kind regards,

Jianhong Zhou

Staff Editor

Journal Requirements:

Additional Editor Comments (if provided):

Reviewers' comments:

Reviewer's Responses to Questions

**Comments to the Author**

1. Does this manuscript meet PLOS Global Public Health’s publication criteria ? Is the manuscript technically sound, and do the data support the conclusions? The manuscript must describe methodologically and ethically rigorous research with conclusions that are appropriately drawn based on the data presented.

Reviewer #1: Yes

2. Has the statistical analysis been performed appropriately and rigorously?

Reviewer #1: Yes

3. Have the authors made all data underlying the findings in their manuscript fully available (please refer to the Data Availability Statement at the start of the manuscript PDF file)?

Reviewer #1: Yes

4. Is the manuscript presented in an intelligible fashion and written in standard English?

Reviewer #1: Yes

5. Review Comments to the Author

Reviewer #1: Dear Authors,

Congratulations on an excellent piece of work. The work is very well written, well researched, and well presented. The findings re-reminded me why I began my own journey within the Gender-WASH space.

I have only a few minor comments to assist in the clarity of several points within the manuscript.

- Results Table 2: It might be valuable to label the community numbers with a letter and number to avoid confusion (e.g H1 for Honduras community 1)

- Results Figure 2: I think the colours for the R-graphs might be attached to countries. If so, could this be added to the caption?

- Discussion: Is there anything more about age or life stage that could be included within the discussion?

- Discussion: You mention some challenges with JMP indicators related to multiple sources for multiple uses, is it worth mentioning the challenges about estimating distance using time? Or do you think that the difficulties in Kenya with animals not going along for the research-water collection negate this?

- Discussion: Is there an opportunity to connect the final paragraph in the discussion back to gender-transformative WASH practices?

- Limitations: You may want to consider a short note about the sampling with regards to drawing generalizable conclusions from these selected communities

- References: Check references for consistency in approach.

6. PLOS authors have the option to publish the peer review history of their article (what does this mean? ). If published, this will include your full peer review and any attached files.

**Do you want your identity to be public for this peer review?** For information about this choice, including consent withdrawal, please see our Privacy Policy .

Reviewer #1: **Yes: ** Jess MacArthur

---

## [Decision Letter · Decision Letter 1]

4 Sep 2025

PGPH-D-25-00317R1

Women’s experiences collecting and accessing water in Guatemala, Honduras, Kenya, and Zimbabwe: A mixed-methods investigation

Dear Dr. Caruso,

Thank you for submitting your manuscript to PLOS Global Public Health. After careful consideration, we feel that it has merit but does not fully meet PLOS Global Public Health’s publication criteria as it currently stands. Therefore, we invite you to submit a revised version of the manuscript that addresses the points raised during the review process.

Please review the reviewer's comments and revise your manuscript accordingly, providing a point-by-point response upon resubmission.

We look forward to receiving your revised manuscript.

Kind regards,

Sarah Jose, Ph.D.

Staff Editor

Journal Requirements:

Additional Editor Comments (if provided):

Reviewer #2:

Reviewers' comments:

Reviewer's Responses to Questions

**Comments to the Author**

1. If the authors have adequately addressed your comments raised in a previous round of review and you feel that this manuscript is now acceptable for publication, you may indicate that here to bypass the “Comments to the Author” section, enter your conflict of interest statement in the “Confidential to Editor” section, and submit your "Accept" recommendation.

Reviewer #2: (No Response)

2. Does this manuscript meet PLOS Global Public Health’s publication criteria ? Is the manuscript technically sound, and do the data support the conclusions? The manuscript must describe methodologically and ethically rigorous research with conclusions that are appropriately drawn based on the data presented.

Reviewer #2: Yes

3. Has the statistical analysis been performed appropriately and rigorously?

Reviewer #2: Yes

4. Have the authors made all data underlying the findings in their manuscript fully available (please refer to the Data Availability Statement at the start of the manuscript PDF file)?

Reviewer #2: Yes

5. Is the manuscript presented in an intelligible fashion and written in standard English?

Reviewer #2: Yes

6. Review Comments to the Author

Reviewer #2: This is a valuable and well-written paper that makes an important contribution to understanding women’s experiences with water collection across diverse contexts. The study design is innovative, and the findings are relevant for both research and practice. Below are suggested revisions to better contextualize the findings and communicate their importance.

1. Throughout: Ensure usage of “data” as plural (e.g., at line 102, it should read “data are,” not “data is”).

2. Abstract: Consider removing the reference to understanding the “gendered burden of this labor.” While the study compellingly documents substantial water-related burdens among women, there is no comparative gender analysis. This does not lessen the contribution of the paper; reframing the aim will better align expectations with the study’s findings.

3. Introduction, line 104: This section could be strengthened by noting that GPS-based approaches to estimating collection time and distance may also have limitations, since the Euclidean distance between a household and its primary water source often does not reflect the route individuals take. See: https://ij-healthgeographics.biomedcentral.com/articles/10.1186/s12942-016-0062-8.

4. Introduction, lines 109–110: Consider strengthening the motivation for this study by emphasizing how improved data on water collection can inform policy and practice, rather than framing the opportunity to address a data gap as inherently meaningful.

5. Methods: Specify the type of mixed-methods design. The description suggests a convergent parallel design.

6. Methods: Ensure reporting aligns with COREQ (https://onlinelibrary.wiley.com/pb-assets/assets/17416612/COREQ_Checklist-1556513515737.pdf) and SRQR (https://journals.lww.com/academicmedicine/fulltext/2014/09000/Standards_for_Reporting_Qualitative_Research__A.21.aspx) standards. Key missing details include number of coders, procedures for ensuring coding reliability, and reflexivity.

7. Methods: Explain the rationale for selecting a multi-site design (e.g., to capture diverse geographic, climatic, and socioeconomic contexts) and whether variation in experiences across sites was expected.

8. Methods: In Kenya, refer to the language as Swahili rather than Kiswahili – the current phrasing would be like saying interviews were conducted in Español rather than Spanish. Review language naming in other contexts to ensure appropriateness.

9. Methods: State whether data collection coincided with rainy or dry seasons. The timing of interviews could strongly influence experiences and should be acknowledged.

10. Methods: Indicate what women were asked to do during the go-along interviews (e.g., to walk to their usual water source and collect water as they typically would). Describe information interviewers were trained to elicit and suggested discussion prompts.

11. Methods, Section 2.3: Consider including the full interview guides and prompts as supplementary material.

12. Methods, line 241: Provide more detail on piloting. Were tools piloted with enumerators or participants? What modifications were made in response to the piloting exercise?

13. Methods, lines 273–274: Include the full list of key themes used in the analysis. Clarify how themes were selected, whether they were updated iteratively as transcripts were read, and how coding was conducted.

14. Methods: Clarify who read the transcripts and created the community profiles. What does it mean that transcripts were “organized by the community”? Provide detail on how profiles were constructed, and when and how community meetings were hosted.

15. Methods, line 296: Clarify whether estimated journey time was recorded before or after water collection. Reflect on how the data collection process itself may have influenced the results. For example, did participants ever choose a nearer water source to reduce the burden on enumerators, potentially leading to shorter travel times? Conversely, could talking with interviewers during the journey have slowed walking pace and resulted in longer recorded times?

16. Results, line 335: Indicate whether classification of improved/unimproved sources was based on JMP criteria or participants’ self-reports.

17. Results, lines 346–347: Clarify whether values represent percentage differences or percentage-point differences.

18. Results, line 354: Specify during which season(s) data were collected. Clarify whether seasonal variation was probed by interviewers or arose spontaneously.

19. Results, lines 340–342: Information on number of trips per day appears later in the manuscript. Consider consolidating these sections for thematic consistency.

20. Results, line 431: State whether distances refer to round-trip or one-way journeys.

21. Results, line 451: Clarify whether terrain challenges were described by participants, observed by enumerators, or interpreted by analysts. This level of transparency should be applied to all descriptive findings.

22. Discussion, line 583: Consider citing an additional relevant exception (see: https://www.journals.uchicago.edu/doi/abs/10.1086/696531).

23. Discussion, lines 577–585: Instead of only emphasizing the novelty of elevation ascent data, discuss its practical significance. What do these findings imply for policy and practice? Should elevation be measured in future studies?

7. PLOS authors have the option to publish the peer review history of their article (what does this mean? ). If published, this will include your full peer review and any attached files.

**Do you want your identity to be public for this peer review?** For information about this choice, including consent withdrawal, please see our Privacy Policy .

Reviewer #2: **Yes: ** Joshua D. Miller

---

## [Decision Letter · Decision Letter 2]

30 Oct 2025

PGPH-D-25-00317R2

Women’s experiences collecting and accessing water in Guatemala, Honduras, Kenya, and Zimbabwe: A mixed-methods investigation

Dear Dr. Caruso,

Thank you for submitting your manuscript to PLOS Global Public Health. After careful consideration, we feel that it has merit but does not fully meet PLOS Global Public Health’s publication criteria as it currently stands. Therefore, we invite you to submit a revised version of the manuscript that addresses the points raised during the review process.

We look forward to receiving your revised manuscript.

Kind regards,

Lina Taing

Academic Editor

Journal Requirements:

Additional Editor Comments (if provided):

Reviewers' comments:

Reviewer's Responses to Questions

**Comments to the Author**

1. If the authors have adequately addressed your comments raised in a previous round of review and you feel that this manuscript is now acceptable for publication, you may indicate that here to bypass the “Comments to the Author” section, enter your conflict of interest statement in the “Confidential to Editor” section, and submit your "Accept" recommendation.

Reviewer #3: (No Response)

2. Does this manuscript meet PLOS Global Public Health’s publication criteria ? Is the manuscript technically sound, and do the data support the conclusions? The manuscript must describe methodologically and ethically rigorous research with conclusions that are appropriately drawn based on the data presented.

Reviewer #3: Yes

3. Has the statistical analysis been performed appropriately and rigorously?

Reviewer #3: (No Response)

4. Have the authors made all data underlying the findings in their manuscript fully available (please refer to the Data Availability Statement at the start of the manuscript PDF file)?

Reviewer #3: Yes

5. Is the manuscript presented in an intelligible fashion and written in standard English?

Reviewer #3: Yes

6. Review Comments to the Author

Reviewer #3: Paper Title: Women’s experiences collecting and accessing water in Guatemala, Honduras, Kenya, and Zimbabwe: A mixed-methods investigation

Paper ID: PGPH-D-25-00317

General Remarks

This manuscript presents a mixed-methods study examining women's water collection experiences in four low-resource countries, combining qualitative insights (go-along interviews and observations) with quantitative measures (smart watches and scales) to quantify time, distance, energy, and other burdens. The research provides real-time evidence, addressing an important gap in understanding the gendered impacts of water insecurity, with implications for policy and interventions. The title is well-worded to capture the central theme of the paper. The abstract is concise and informative, the introduction provides a solid literature review, and the methods section details unique approaches using technology for precise measurements.

The strengths of the article include the multi-country scope, integration of mixed methods, and focus on underrepresented aspects like cognitive burden and energy expenditure – providing research-driven evidence to overlooked but important nuances. The collaboration with local partners also enhances contextual relevance. Other observations and corrections are highlighted below under respective subsections:

Abstract

Line 41: "Mean caloric expenditure was 231cal (range: 36 (Guatemala) – 952 (Zimbabwe))." Units could be clarified for scientific accuracy. "Cal" is used, but in nutrition/energy contexts, it's typically "kcal" (kilocalories) to avoid confusion with small calories. Smartwatch data may likely estimate kcal. Kindly re-check.

Line 30: Replace “experiences is” with “experiences are”

Lines 38–42: The ranges (e.g., for time: 13 min in Guatemala to 287 min in Kenya) imply overall minimum-maximum values for specific countries. Clarify if these are extrema across all participants or per-country mins/maxes.

Keywords: There are no keywords. Include about 4 – 6 keywords, arranged in alphabetical order

Introduction

Line 57: Change ‘responsibility’ to ‘responsibilities’

Line 67: Change ‘injury’ to ‘injuries’

Line 85: Add ‘a’ before ‘…more precise’

Line 94: Write ‘roundtrip’ to ‘round-trip’

Line 102 – 103: Write ‘Traveled’ as ‘Travelled’

Methods

Line 134: Figure 1 is missing

Line 150: Change ‘sample of individuals who were eligible’ to ‘sample of eligible individuals’

Line 187: Change ‘was’ to ‘were’

Line 208: Address missing Figure 1

Other specific remarks:

1. In Methods (Line 155; Page 8, Line 155): "In Guatemala, 95% of the population (91% rural; 98% urban) has at least basic drinking water service (i.e., from an improved source requiring no more than 30 minutes roundtrip for collection) (4)." You may revise as "In Guatemala, 95% of the population (91% rural; 98% urban) has at least basic drinking water service, defined as access to an improved source requiring no more than 30 minutes round-trip for collection per WHO/UNICEF JMP (4)."

2. The use of convenience sampling [Line 150] and purposive community selection by World Vision [Line 141] introduces potential selection bias, as communities were chosen based on SWSW program engagement. This limits the representativeness of findings across broader populations. Can you include a sensitivity analysis or discuss how these biases might affect external validity? You may consider random sampling in future studies.

3. Provide additional descriptive statistics with measures of variability (e.g., Standard Deviation and Standard error) for key outcomes like time and distance. Also, consider inferential statistics to compare countries, as raw means/ranges alone limit interpretation. For instance, quantitative results (e.g., mean caloric expenditure of 231 cal [Line 41]) lack measures of variability (e.g., standard deviation) or statistical tests to compare across countries. Describe MEANs with standard errors or conduct ANOVA to test country-specific variations.

4. Include example quotes in the results to illustrate themes (e.g., mental burden). Also, you can use a framework like thematic analysis to structure findings. Also, while themes like risks from terrain and animals [Line 37] are noted, the absence of direct quotes or thematic coding limits the richness of experiential data. You can incorporate representative quotes and a thematic framework (e.g., grounded theory) to enhance qualitative rigour.

5. Discuss how dry-season data collection may overestimate burdens - data collection during the dry season in Zimbabwe [Line 186] and variable seasons across countries may skew results (e.g., longer distances in Kenya, 15.8 km [Line 40]), as water scarcity intensifies burdens [Line 80]. Can you analyze seasonal effects separately and leverage longitudinal data to assess variability?

6. Acknowledge the non-representative sample and suggest future population-based studies, if any.

7. Discuss how risks (e.g., accompanying women on journeys) were mitigated and how data privacy was ensured for the smart watch GPS.

8. Provide the DOI number for the listed references

7. PLOS authors have the option to publish the peer review history of their article (what does this mean? ). If published, this will include your full peer review and any attached files.

**Do you want your identity to be public for this peer review?** For information about this choice, including consent withdrawal, please see our Privacy Policy .

Reviewer #3: No

 Figure Resubmissions:

---

## [Editor Report · Decision Letter 3]

12 Nov 2025

Women’s experiences collecting and accessing water in Guatemala, Honduras, Kenya, and Zimbabwe: A mixed-methods investigation

PGPH-D-25-00317R3

Dear Dr. Caruso,

We are pleased to inform you that your manuscript 'Women’s experiences collecting and accessing water in Guatemala, Honduras, Kenya, and Zimbabwe: A mixed-methods investigation' has been provisionally accepted for publication in PLOS Global Public Health.

Best regards,

Lina Taing

Academic Editor

This is a novel and impactful contribution to understanding the true costs of inadequate water access for women globally. I particularly appreciate how your analysis moves beyond conventional measures of time or distance to incorporate the physical and physiological dimensions of water collection—such as terrain, weight carried, and cumulative health strain. This multidimensional framing powerfully captures the embodied burden of water insecurity and makes visible the depletion of women’s energy, opportunities, and well-being that often remains unquantified. By grounding your analysis in women’s lived experiences, you compellingly highlight the intrinsic value of women’s labor and health, challenging the narrow, instrumental approaches that dominate much of WASH research. Your findings make a strong case for water programs that acknowledge and reduce unnecessary burdens, prevent harm, and advance genuinely gender-aware solutions.